# IRBridge: Solving Image Restoration Bridge with Pre-trained Generative Diffusion Models

**Hanting Wang** [† 1] **Tao Jin** [† 1] **Wang Lin** [1] **Shulei Wang** [1] **Hai Huang** [1] **Shengpeng Ji** [1] **Zhou Zhao** [* 1]

## Abstract

Bridge models in image restoration construct a diffusion process from degraded to clear images. However, existing methods typically require training a bridge model from scratch for each specific type of degradation, resulting in high computational costs and limited performance. This work aims to efficiently leverage pretrained generative priors within existing image restoration bridges to eliminate this requirement. The main challenge is that standard generative models are typically designed for a diffusion process that starts from pure noise, while restoration tasks begin with low-quality images, resulting in a mismatch in the state distributions between the two processes. To address this challenge, we propose a transition equation that bridges two diffusion processes with the same endpoint distribution. Based on this, we introduce the ***IRBridge*** framework, which enables the direct utilization of generative models within image restoration bridges, offering a more flexible and adaptable approach to image restoration. Extensive experiments on six image restoration tasks demonstrate that IRBridge efficiently integrates generative priors, resulting in improved robustness and generalization performance. Code will be available at GitHub.

## 1. Introduction

Image restoration seeks to reconstruct high-quality images from degraded inputs affected by factors such as rain, fog, blur, or noise. As a longstanding fundamental problem, it has been extensively studied over the past decades. Mainstream learning-based methods typically adopt an end-to-end supervised paradigm, training regression models for pixel-level reconstruction. While these approaches achieve

---

[†]Equal contribution [*]Corresponding author [1]Zhejiang University. Correspondence to: Zhou Zhao <zhaozhou@zju.edu.cn>.

*Proceedings of the $42^{nd}$ International Conference on Machine Learning*, Vancouver, Canada. PMLR 267, 2025. Copyright 2025 by the author(s).

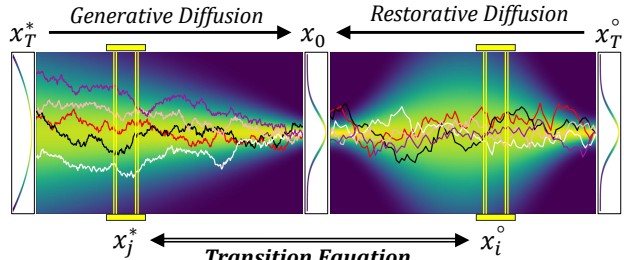

Figure 1: Our core contribution lies in bridging the transition between two diffusion processes with the same endpoint distribution, enabling the leverage of pretrained generative priors in image restoration bridges.

notable performance, studies (Delbracio & Milanfar, 2023) reveal that their learning objective often converges to the mean of multiple plausible solutions, limiting their ability to recover fine details such as intricate textures.

In contrast, advancements in generative modeling, particularly within the family of diffusion models (Sohl-Dickstein et al., 2015; Ho et al., 2020; Song et al., 2020a;b), have demonstrated remarkable performance. These models are capable of generating satisfying, visually detailed image samples, attracting attempts (Zhu et al., 2023b; Chung et al., 2022; Fei et al., 2023; Gou et al., 2024) to explore their potential in image restoration tasks. However, most of these methods follow the standard diffusion process, diffusing images into pure noise and using degraded images as the condition for generation, which often requires integrating prior knowledge of the degradation process.

To model the transition from low-quality images to high-quality images more intuitively, *bridge models* (Liu et al., 2022a; De Bortoli et al., 2021; Zhou et al., 2023) have garnered increasing attention, as they model the stochastic process between two distributions or fixed points. Compared to generation processes initiated from noise, these methods (Delbracio & Milanfar, 2023; Luo et al., 2023; Yue et al., 2023a) better align with the physical dynamics of image restoration, offering distinct theoretical advantages (Delbracio & Milanfar, 2023; Yue et al., 2023a).

However, these approaches invariably require training a model from scratch for each type of degradation, limited by

dataset scale and model capacity. Given the consistency in target distributions between image generation models and image restoration bridge models (*i.e.*, both regard the distribution of high-quality images as their target distribution), a hopeful avenue is to leverage their powerful generative priors in the context of image restoration bridges. To achieve this, a key challenge lies in the differences between the two diffusion processes. The mismatch between the distributions of intermediate states in the process flows prevents the pretrained generative model from directly handling the states of the image restoration bridge.

To address this challenge, we propose a transition equation (Section 3.1) that bridges two stochastic processes sharing the same endpoint distribution, enabling transitions between their states. We analyze the boundary conditions (Section 3.2) of the transition equation and define *forward transition* and *reverse transition* accordingly. Building on the above concepts, we propose the **IRBridge** framework (Section 3.3), which enables the application of pre-trained generative diffusion models to solve the image restoration bridge problem. Additionally, we explore inference-time hyperparameter strategies (Section 5.1) for various tasks to optimize performance. Our framework eliminates the need to train restorative bridge models from scratch for specific types of degradation and reduces training costs while enhancing model capacity and generalization performance.

Our main contributions can be summarized as follows:

1. We propose a transition equation that bridges two diffusion processes with identical endpoint distributions, enabling state transitions between an image restoration bridge and a generative diffusion model.

2. We explore the boundary conditions of the proposed transition equation, leading to a novel image restoration framework, IRBridge, which effectively integrates pretrained generative priors into the image restoration bridge.

3. The proposed framework eliminates the need to train a separate image restoration bridge model from scratch for each specific type of degradation. Additionally, it provides greater flexibility by allowing the customization of inference hyperparameters for different tasks to optimize performance.

4. Extensive experiments on 6 mainstream image restoration tasks demonstrate that the integrated generative priors significantly enhance the robustness and generalization performance of the image restoration bridge model.

## 2. Preliminaries

### 2.1. Generative Diffusion Models

Generative diffusion models (Ho et al., 2020; Song & Ermon, 2019; Rombach et al., 2022) define a forward noising process that gradually perturbs the data distribution into a known tractable prior by progressively injecting noise. A corresponding reverse process is then learned to generate data by reversing this transformation. From the perspective of stochastic differential equations (SDEs), generative diffusion models can be formulated (Song et al., 2020b) as discretizations of different types of SDEs, typically categorized into variance-preserving (VP) and variance-exploding (VE) formulations. In the case of DDPM(Ho et al., 2020), its forward diffusion process is defined by a Markov chain. At any timestep $t$, the state $x_t$ has a closed-form conditional distribution:

$$p(x_t) = \mathcal{N}(x_t; \sqrt{\bar{\alpha}_t}x_0, (1 - \bar{\alpha}_t)\mathbf{I}) \qquad (1)$$

where $\bar{\alpha}_t$ are predefined coefficients. A denoising network $\epsilon_\theta(x_t, t)$ is trained to predict the added noise $\epsilon \sim \mathcal{N}(0, \mathbf{I})$, and the original sample $x_0$ can then be estimated as:

$$\hat{x}_0 = \frac{x_t - \sqrt{1 - \bar{\alpha}_t}\epsilon_\theta(x_t, t)}{\sqrt{\bar{\alpha}_t}} \qquad (2)$$

### 2.2. Image Restoration Bridge Models

We primarily discuss two types of image restoration bridges: IR-SDE(Luo et al., 2023) and GOUB(Yue et al., 2023a). IR-SDE employs a mean-reverting SDE that transforms a high-quality image $x_{hq}$ into its degraded counterpart $x_{lq}$, treating the latter as the mean of the terminal Gaussian distribution. The employed SDE is formulated as:

$$dx_t = \theta_t(\mu - x_t)dt + g_t d\mathbf{w}_t \qquad (3)$$

where $\mathbf{w}_t$ is standard Brownian motion and $\mu := x_{lq}$. Suppose that the SDE coefficients satisfy $g_t^2/\theta_t := 2\lambda^2$ for all times $t$, the solution(Luo et al., 2023) to the SDE determined in Eq 3 is:

$$p(x_t) = \mathcal{N}(x_t; m_t, v_t\mathbf{I})$$
$$m_t = \mu + (x_0 - \mu)e^{-\bar{\theta}_t} \qquad (4)$$
$$v_t = \lambda^2(1 - e^{-2\bar{\theta}_t})$$

where $\bar{\theta}_t = \int_0^t \theta_z dz$. Similar to diffusion models, IR-SDE restores images by training a neural network to estimate the score function of the reverse SDE corresponding to Eq 3. GOUB(Yue et al., 2023a) further incorporates the Doob's $h$-transform (Särkkä & Solin, 2019) into the SDE described by Eq 3, eliminating the noise at the diffusion endpoint:

$$dx_t = (\theta_t + g_t^2 \frac{e^{-2\bar{\theta}_{t:T}}}{\bar{\sigma}_{t:T}^2})(x_T - x_t)dt + g_t d\mathbf{w}_t \qquad (5)$$

where $\bar{\sigma}_{s:t}^2 := \frac{g_t^2}{2\theta_t}(1 - e^{-2\bar{\theta}_{s:t}})$ and $x_T := x_{lq}$. Correspondingly, the forward transition is given by:

$$p(x_t) = \mathcal{N}(x_t; \bar{m}_t, \bar{v}_t \mathbf{I}),$$

$$\bar{m}_t = e^{-\bar{\theta}t}\frac{\bar{\sigma}_{t:T}^2}{\bar{\sigma}_T^2}x_0 + \left[(1 - e^{-\bar{\theta}t})\frac{\bar{\sigma}_{t:T}^2}{\bar{\sigma}_T^2} + e^{-2\bar{\theta}t:T}\frac{\bar{\sigma}_t^2}{\bar{\sigma}_T^2}\right]x_T,$$

$$\bar{v}_t = \bar{\sigma}_t^2 \frac{\bar{\sigma}_{t:T}^2}{\bar{\sigma}_T^2},$$

$$\tag{6}$$

GOUB can be viewed as imposing boundary conditions on the generalized Ornstein-Uhlenbeck process described in Eq 3, by specifying fixed endpoints for the stochastic trajectory.

# 3. Method

Our goal is to effectively integrate pre-trained generative models into image restoration bridges. A significant challenge arises from the differing diffusion processes defined by image generation models and image restoration bridges. This discrepancy causes their states to follow distinct distributions, preventing pre-trained generative models from directly handling the states of image restoration bridges. We first introduce the proposed transition equation, which bridges two diffusion processes with the same endpoint distributions, addressing this challenge. Then, we discuss the validity conditions for this equation, followed by the introduction of the proposed IRBridge framework.

## 3.1. Transition Equation

Assume that high-quality image samples $x_0$ follow the data distribution $P_{data}$. It can be concluded that the endpoints of the diffusion processes defined by generative diffusion models and image restoration bridge models share the same distribution $P_{data}$. For simplicity, we rewrite Eq (1, 4, 6) into the reparameterized form $x_t = f_t x_0 + b_t + \sigma_t \epsilon$. This form indicates that the state $x_t$ at any given time $t$ can be expressed as a linear combination of the image sample $x_0$, a known offset term $b_t$, and standard noise $\epsilon$. Specifically, in the case of DDPM, the offset term $b_t$ is always zero, whereas for IR-SDE and GOUB, it is a known coefficient related to the degraded image $x_{lq}$. We use superscripts to differentiate them: generally, $x_i^\circ$ represents the state at time $i$ in the image restorative bridge, and $x_j^*$ represents the state at time $j$ in generative diffusion. Typically, both generative models and existing bridge models for image restoration adopt similar linear Gaussian paths. We present reparameterizations of different categories of diffusion models in the Appendix B (Table 2) to illustrate this insight.

Since the diffusion processes are completely independent, the states of the two processes are conditionally independent given a sample $x_0 \sim P_{data}$. This conditional independence leads to the following transition equation:

**Proposition 3.1.** *Given two diffusion processes with the same endpoint distribution $P_{data}$ and a sample $x_0 \sim P_{data}$, the transition between their states $x_i^\circ, x_j^*$ at any timestep $i, j$ can be expressed as:*

$$x_j^* = \alpha \cdot x_i^\circ + \beta \cdot x_0 + \gamma + \sigma\epsilon,$$

$$\alpha = \sqrt{\frac{(\sigma_j^*)^2 - \sigma^2}{(\sigma_i^\circ)^2}},$$

$$\beta = f_j^* - \sqrt{\frac{(\sigma_j^*)^2 - \sigma^2}{(\sigma_i^\circ)^2}} \cdot f_i^\circ,$$

$$\gamma = b_j^* - \sqrt{\frac{(\sigma_j^*)^2 - \sigma^2}{(\sigma_i^\circ)^2}} \cdot b_i^\circ,$$

$$\tag{7}$$

*where $\alpha$ and $\beta$ represent the coefficients of $x_i^\circ$ and $x_0$,respectively, while $\gamma$ denotes a known drift. The noise coefficient $\sigma$ can be treated as a controllable variable. Additionally, Eq 7 is actually a reparameterization. Therefore, the condition for its validity is $\alpha, \beta, \sigma \geq 0$, which leads to $\max\{0, [(\sigma_j^*)^2 - (\frac{f_j^* \cdot \sigma_i^\circ}{f_i^\circ})^2]\} \leq \sigma^2 \leq (\sigma_j^*)^2$.*

Through the Proposition 3.1, the transition between the states of two diffusion processes can be achieved. Therefore, by transforming the state of an image restoration bridge onto the diffusion trajectory of a generative model, the pre-trained denoising network can be applied to estimate the initial state $x_0$, thereby enabling the iterative process to proceed.

However, this transformation necessitates a pre-estimated $x_0$, which is inaccessible during actual inference. Fortunately, through further discussion of the boundary conditions in Eq 7, we find that appropriate choices of hyperparameters can help us circumvent the need for prior estimate.

## 3.2. Boundary Condition

We treat the noise term $\sigma$ in Eq 7 as a manually determined coefficient for the transition. When $\sigma$ reaches its maximum value, the coefficient $\alpha$ becomes 0, and Eq 7 degenerates into the general form of the forward diffusion process. However, our primary focus is on the scenario where $\sigma$ attains its minimum value. In this case, the transition equation exhibits minimal dependence on $x_0$ and incorporates the smallest noise term.

**Critical Timestep.** Given the timestep $i$ of the source state $x_i^\circ$, we refer to the timestep $j$ that satisfies $(\frac{f_i^\circ}{f_j^*})^2 = (\frac{\sigma_j^*}{\sigma_i^\circ})^2$ as the *Critial Timestep* $\tilde{t}_i$. When $j = \tilde{t}_i$ and $\sigma$ reaches its minimum value of 0, the coefficient $\beta$ of $x_0$ in Eq 7 also becomes 0. The critical timestep is essentially the smallest timestep that allows the coefficient $\beta$ to take the value of 0.

Note that the value of $\tilde{t}_i$ depends solely on the diffusion coefficients defined by the two diffusion processes, which

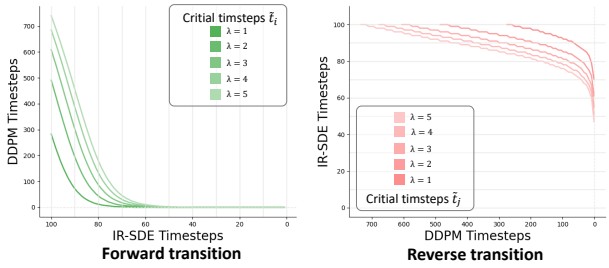

**Forward transition**       **Reverse transition**

Figure 2: An example of critical timesteps: the generative diffusion model is Stable Diffusion v1-5 (Rombach et al., 2022), while the restorative diffusion model is IR-SDE (Luo et al., 2023).

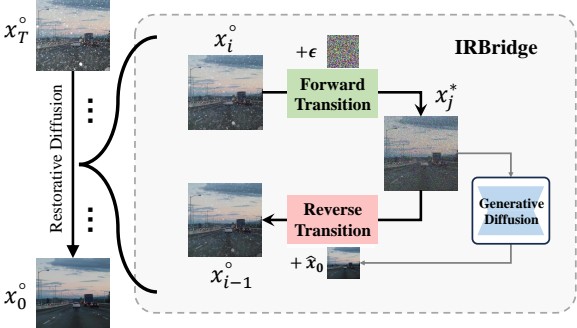

Figure 3: Overview of the proposed IRBridge framework. This framework enables the direct application of pre-trained models in image restoration bridges, effectively leveraging their powerful priors to achieve image restoration.

provides guidance for selecting the parameter $\lambda$ (in Eq (3, 5)) of image restoration bridge models. Figure 2 illustrates an example of Stable Diffusion v1-5 and IR-SDE. In general, a smaller value of $\lambda$ result in smaller critical timesteps. Typically, we choose $\lambda$ in such a way that the critical timestep $\tilde{t}_i$ falls within the range of the pre-trained DDPM timesteps.

**Forward Transition.** If we select $j > \tilde{t}_i$ (*i.e.*, above the line in Figure 2 left) such that $(\sigma_j^*)^2 - (\frac{f_j^* \cdot \sigma_i^\circ}{f_i^\circ})^2 > 0$, then by choosing the smallest $\sigma = \sqrt{(\sigma_j^*)^2 - (\frac{f_j^* \cdot \sigma_i^\circ}{f_i^\circ})^2}$, we can ensure that the coefficient $\beta$ of $x_0$ in Eq 7 becomes 0. At this point, Eq 7 simplifies to a process of adding noise to the source state $x_i^\circ$ independently of $x_0$. This implies that we can convert the state of the image restoration bridge into the state of DDPM without requiring a prior estimate of $x_0$.

Following the DDPM convention, we refer to this process as the *forward transition*. It is worth noting that no additional constraints are placed on the selection of $j$. We will further illustrate that the strategy for selecting $j$ primarily depends on the characteristics of the specific task (Section 5.1).

**Reverse Transition.** Once we have an estimate of $x_0$, we can similarly apply the transition equation to convert the state $x_j^*$ of DDPM back into the state $x_{i-1}^\circ$ of the image restoration bridge, thereby supporting the next iteration. We invert the roles of the restorative diffusion and the generative diffusion in Eq 7 and ensure that the next timestep $(i-1)$ is less than the critical timestep $\tilde{t}_j$ (*i.e.*, below the line in Figure 2 right) such that $(\sigma_j^*)^2 - (\frac{f_j^* \cdot \sigma_i^\circ}{f_i^\circ})^2 < 0$. At this point, if we choose the minimum value of $\sigma = 0$, the coefficient of $x_0$ in Eq 7 remains $\beta \geq 0$. Since the noise term is zero, Eq 7 simplifies to a deterministic transformation dependent on $x_0$. We refer to this transition as the *reverse transition*.

### 3.3. IRBridge Framework

Building on the preceding concepts, we propose a novel image restoration framework, *IRBridge*, which leverages pre-trained generative models to solve image restoration

bridges. As illustrated in Figure 3, IRBridge first applies a forward transition to convert the state $x_i^\circ$ of the restoration bridge to a state of DDPM trajectory, enabling the pre-trained generative model to estimate the initial sample $x_0$. A subsequent reverse transition then transforms this state into $x_{i-1}^\circ$, facilitating the next iterative step. By repeatedly performing these transitions, IRBridge achieves image restoration via conditional generative models (discussed in Section 5.2), effectively harnessing their generative priors.

It is important to emphasize that Eq 7 describes the transition bridging two Gaussian conditional probability paths, requiring the noise term $\sigma_i^\circ$ in the forward process to be non-zero. In certain cases, such as diffusion bridge models like GOUB (Yue et al., 2023a), which model a stochastic process between two fixed points, the noise coefficient at its initial timestep is typically zero. This renders the transition equation invalid. To address this issue, we directly skip the initial timestep in this situation during inference. Experiments (Section 4) show that this simple strategy does not negatively impact performance.

IRBridge offers flexible parameter selection, enabling the adjustment of the inference process by selecting appropriate hyperparameters for specific task (Section 5.1). It is also noteworthy that reverse transition is not exclusive, leveraging the reverse process of the original image restoration bridge represents a viable alternative (Section 5.4).

The proposed framework obviates the need to train a distinct image restoration bridge model for each type of degradation from scratch and avoids explicitly modeling the degradation process. Instead, it only requires a conditional generative model to solve the image restoration bridge. Since IRBridge leverages pretrained generative diffusion models, which are typically trained on large-scale image datasets, it benefits from improved robustness and generalization performance compared to bridge models trained from scratch.

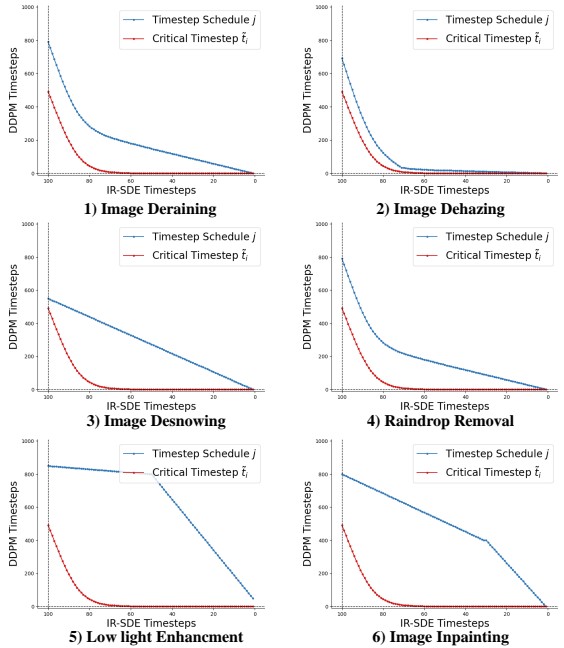

Figure 4: Timestep schedules selected for different image restoration tasks (showing only the cases for IR-SDE). The red lines represent the critical timesteps, while the blue lines indicate the chosen DDPM timesteps.

# 4. Experiments

## 4.1. Settings

We utilize Stable Diffusion v1-5 (Rombach et al., 2022) as the pretrained generative diffusion model for IRBridge. For the image restoration bridge, we choose IR-SDE (Luo et al., 2023) and GOUB (Yue et al., 2023a) for experiments, as they use distribution and fixed points, respectively, as the endpoints of their diffusion processes. Unless otherwise specified, the hyperparameter $\lambda$ in Eq (4, 6) of image restoration bridge models is set to 2. To introduce conditions into the generative model, we employ ControlNet (Zhang et al., 2023), which is known for its efficiency in integrating guidance conditions without compromising the model's original capabilities. For all tasks, degraded images are directly used as conditions for ControlNet. We directly use empty text as the textual conditions and classifier-free guidance is not employed. For more details, please refer to Appendix E.

Our experiments cover 6 major image restoration tasks, including **image deraining, dehazing, desnowing, raindrop removal, low-light enhancement, and image inpainting**. These degradations stem from diverse causes and vary in their impact, influencing attributes such as layout, illumination, and texture. Figure 4 illustrates the timestep schedules adopted for different image restoration tasks. It should be noted that these timestep schedules are not theoretically optimal. We will further discuss this in Section 5.1.

## 4.2. Results and Analysis

Table 1 present the quantitative comparison results of our IRBridge with other methods on the aforementioned tasks. We employ PSNR, SSIM, LPIPS, and FID as evaluation metrics. PSNR quantifies pixel-level discrepancies, SSIM assesses structural similarities (Wang et al., 2004), and LPIPS (Zhang et al., 2018) measures perceptual similarity based on high-level semantic features. In contrast, FID (Heusel et al., 2017) evaluates the distance between the distributions of generated and real images, emphasizing overall image realism rather than individual pixel accuracy. Consistent with prior studies (Luo et al., 2023; Yue et al., 2023a), we report PSNR and SSIM scores specifically on the Y channel within the YCbCr color space. To eliminate potential interference introduced by the VAE employed by pre-trained stable diffusion model, these metrics are computed using label images decoded by the same VAE.

**Outperforming bridge models trained from scratch.** As shown in Table 1, although IRBridge is not directly train models to estimate the conditional score of image restoration bridges, it still outperforms image restoration bridge models trained from scratch (ResShift(Yue et al., 2023b), IRSDE (Luo et al., 2023), GOUB (Yue et al., 2023a), RDDM(Liu et al., 2024), DiffUIR(Zheng et al., 2024)) on most image restoration tasks. These results, particularly the comprehensive improvement in FID scores, highlight the advantages brought by integrating pretrained generative priors into IR-Bridge. Compared to these bridge models trained from scratch on limited-scale image restoration datasets, generative models learned through pretraining on large-scale image datasets provide stronger feature representation capabilities, leading to enhanced performance. Moreover, leveraging pretrained models also significantly reduces training burden. For example, training GOUB from scratch takes about 2.5 days on an Nvidia 3090 GPU (see Appendix E of (Yue et al., 2023a)), while the ControlNet we employed achieves similar results in just about 1 day.

**Enhanced generalization performance.** Figure 7 presents a visual comparison between IRBridge and bridge model trained from scratch (GOUB) on inpainting tasks for various other scenes, including both indoor and outdoor images. All models are trained exclusively on **facial image data**. As observed in the highlighted regions of the figure, IRBridge consistently maintains superior visual quality, whereas GOUB fails to fully inpaint the missing areas and delivers unsatisfactory details (*e.g.*, noticeable blurred regions in the staircase area and book pages). These results intuitively demonstrate that the pretrained generative priors endow IRBridge with the ability to generalize to unseen data distributions during training. Moreover, the results also indicate that the employed ControlNet primarily learns to utilize existing guidance information in the low-quality

Table 1: Quantitative comparison across 6 mainstream image restoration tasks. We use 'BM' to denote restorative bridges, 'RM' to represent regression-based methods, and 'DGP' to denote methods leveraging generative diffusion priors. All results are evaluated at a resolution of 512. We choose IR-SDE and GOUB as the diffusion processes during inference.

| Method | Type | Deraining | | | | Method | Type | Dehazing | | | |
| | | PSNR↑ | SSIM↑ | LPIPS↓ | FID↓ | | | PSNR↑ | SSIM↑ | LPIPS↓ | FID↓ |
|---|---|---|---|---|---|---|---|---|---|---|---|
| ResShift[†] | BM | 27.54 | 0.8305 | 0.054 | 30.87 | ResShift[†] | BM | 24.93 | 0.8107 | 0.097 | 17.89 |
| IR-SDE | BM | 31.65 | 0.9041 | 0.047 | 18.64 | TAO | DGP | 20.07 | 0.7980 | 0.105 | 21.84 |
| GOUB | BM | 31.96 | 0.9028 | 0.046 | 18.14 | WGWS-Net[†] | RM | 26.40 | 0.9015 | 0.051 | 14.15 |
| RDDM[†] | BM | 28.97 | 0.9050 | 0.058 | 28.80 | RDDM[†] | BM | 25.33 | 0.8773 | 0.088 | 20.87 |
| DiffUIR[†] | BM | 29.87 | 0.9619 | 0.029 | 10.98 | DiffUIR[†] | BM | 25.99 | 0.8669 | 0.057 | 24.87 |
| IRBridge-IRSDE | - | 28.27 | 0.9583 | 0.032 | 9.21 | IRBridge-IRSDE | - | 26.85 | 0.8755 | 0.065 | 15.73 |
| IRBridge-GOUB | - | 30.39 | 0.9639 | 0.027 | 8.64 | IRBridge-GOUB | - | 26.37 | 0.8752 | 0.067 | 15.82 |

| Method | Type | Desnowing | | | | Method | Type | Raindrop Removal | | | |
| | | PSNR↑ | SSIM↑ | LPIPS↓ | FID↓ | | | PSNR↑ | SSIM↑ | LPIPS↓ | FID↓ |
|---|---|---|---|---|---|---|---|---|---|---|---|
| ResShift[†] | BM | 24.14 | 0.7551 | 0.1697 | 29.87 | ResShift[†] | BM | 25.78 | 0.7958 | 0.204 | 58.79 |
| WGWS-Net[†] | RM | 20.34 | 0.7023 | 0.2424 | 35.87 | WGWS-Net[†] | RM | 28.28 | 0.8686 | 0.102 | 54.09 |
| WeathderDiff | BM | 21.79 | 0.6721 | 0.2286 | 33.96 | WeathderDiff | BM | 27.06 | 0.8473 | 0.089 | 49.06 |
| RDDM[†] | BM | 24.81 | 0.7810 | 0.1634 | 31.84 | RDDM[†] | BM | 27.07 | 0.8048 | 0.102 | 51.86 |
| DiffUIR[†] | BM | 25.07 | 0.7812 | 0.1593 | 22.89 | DiffUIR[†] | BM | 26.88 | 0.8247 | 0.114 | 55.84 |
| IRBridge-IRSDE | - | 25.78 | 0.7832 | 0.1574 | 13.17 | IRBridge-IRSDE | - | 27.08 | 0.8165 | 0.098 | 48.39 |
| IRBridge-GOUB | - | 25.87 | 0.7813 | 0.1569 | 10.38 | IRBridge-GOUB | - | 26.91 | 0.8132 | 0.098 | 49.15 |

| Method | Type | Low light Enhancement | | | | Method | Type | Image Inpainting | | | |
| | | PSNR↑ | SSIM↑ | LPIPS↓ | FID↓ | | | PSNR↑ | SSIM↑ | LPIPS↓ | FID↓ |
|---|---|---|---|---|---|---|---|---|---|---|---|
| ResShift[†] | BM | 22.98 | 0.7982 | 0.119 | 50.54 | DDRM | DGP | 27.16 | 0.8893 | 0.089 | 37.02 |
| GDP | DGP | 14.55 | 0.6432 | 0.217 | 48.67 | IR-SDE$_{CNN}$ | RM | 29.22 | 0.9218 | 0.065 | 38.35 |
| TAO | DGP | 16.57 | 0.7823 | 0.186 | 45.74 | IR-SDE | BM | 28.37 | 0.9166 | 0.046 | 25.13 |
| RDDM[†] | BM | 21.89 | 0.8251 | 0.124 | 46.70 | GOUB | BM | 28.98 | 0.9067 | 0.037 | 4.30 |
| DiffUIR[†] | BM | 24.29 | 0.8853 | 0.100 | 49.88 | PromptIR | RM | 30.22 | 0.9180 | 0.068 | 32.69 |
| IRBridge-IRSDE | - | 25.52 | 0.9232 | 0.059 | 36.26 | IRBridge-IRSDE | - | 30.41 | 0.9159 | 0.066 | 3.34 |
| IRBridge-GOUB | - | 25.59 | 0.9236 | 0.059 | 36.11 | IRBridge-GOUB | - | 30.42 | 0.9158 | 0.067 | 3.37 |

image rather than the process of removing degradation.

In addition, we provide a quantitative comparison between IRBridge and other methods in real-world degradation scenarios in the Appendix G. The results of no-reference image quality assessment (NR-IQA) indicate that the proposed IRBridge achieves better performance. These results demonstrate that incorporating pretrained generative priors enhances the generalization ability of image restoration bridges across different image domains and under complex real-world degradations.

**Comparison with methods utilizing pretrained generative priors.** We also compare IRBridge with other methods (GDP (Fei et al., 2023), TAO (Gou et al., 2024)) that leverage pretrained generative priors but follow the standard generative diffusion paradigm. As shown in Table 1, IRBridge consistently outperforms these methods across all metrics. Unlike these methods that start their diffusion process from pure noise, IRBridge initiates its iterative process directly from the degraded image. This allows IRBridge to anchor itself to the degraded image throughout its inference process, ensuring better data consistency. We will further

discuss the impact of the diffusion process in Section 5.3.

**Comparison with regression-based image restoration methods.** As shown in Table 1, although IRBridge is not explicitly trained for image restoration tasks, it achieves performance comparable to the regression-based model WG-WSNet (Zhu et al., 2023a), which follows an end-to-end supervised learning paradigm. Remarkably, IRBridge even outperforms WGWSNet in the image desnowing task. It is also important to highlight that the current results do not reflect the theoretical performance limits of IRBridge (see Section 5.1). These findings underscore the effectiveness of IRBridge in leveraging conditional generative models for image restoration

## 5. Discussions

### 5.1. Timestep Selection Strategy

The timestep selection strategy for the forward transition process in IRBridge is primarily task-specific and relies on empirical choices. In this section, we provide a high-level analysis of this strategy.

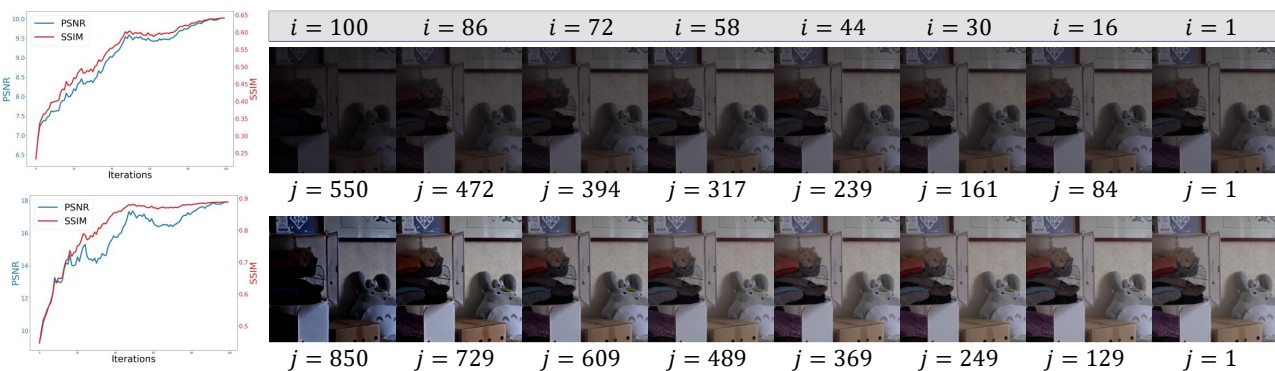

Figure 5: The impact of timestep scheduling on the model's prediction $x_0$. We simply select a linear timestep schedule, with the top row showing the schedule for lower values ($j \in [550, 1]$) and the bottom row showing the schedule for higher values ($j \in [850, 1]$). On the right, the changes in PSNR and SSIM with iterations are shown synchronously.

**DDPM Timestep Setting.**

Figure 5 illustrates the impact of different timestep schedules on the low-light enhancement task, presenting both PSNR/SSIM curves and corresponding visual results. We evaluate two simple linear scheduling strategies. The top and bottom rows of the figure correspond to timestep ranges $j \in [550, 1]$ and $j \in [850, 1]$, respectively, showcasing intermediate model predictions $\hat{x}_0$ across iterations along with their associated performance curves.

The results indicate that using fewer timesteps leads to initial state estimations with richer visual details, but the images remain underexposed until the final stages. In contrast, schedules with larger timesteps (*e.g.*,$j = 850$) produce less accurate initial predictions but progressively improve brightness and detail, ultimately achieving better overall enhancement performance.

We explain that low-light images typically have a lower mean compared to regular images, and the low-level noise added by a lower $j$ during the forward transition is insufficient to disrupt the statistical properties of the degraded distribution. As a result, the final output still suffers from poor lighting conditions. Therefore, we recommend using higher timesteps $j$ for low-quality images with more severe degradation.

**Temporal Dynamic.** Simply using linear scheduling does not achieve satisfactory performance across all tasks. Figure 6 provides key insights by visualizing the coefficient curves for both forward and reverse transitions. It can be observed that the coefficient $\beta$ of $x_0$ in the reverse transition (blue line in Figure 6 right) rises rapidly as the iterations progress. Therefore, we argue that maintaining a higher timestep $j$ in the early stages is appropriate, as the estimated $\hat{x}_0$ has a relatively small influence on $x_{i-1}^\circ$ but provides a solid initialization. In the middle stage, the timestep $j$ should be adjusted based on task characteristics: for tasks with severe degradation (*e.g.*, low-light enhancement), a higher

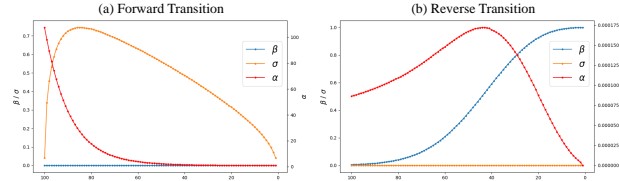

Figure 6: The variation of the forward/reverse transition coefficients ($\alpha, \beta, \sigma$ in Eq 7) with iterations under the linear timestep scheduling.

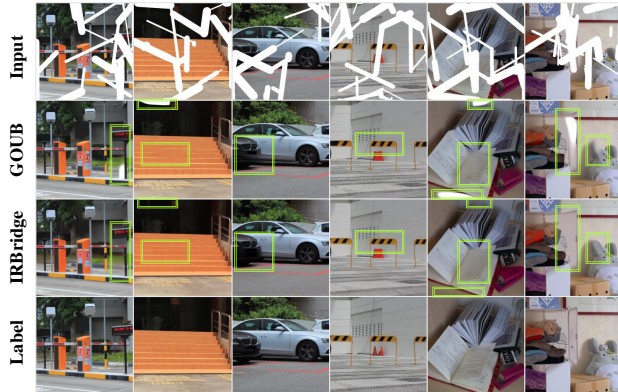

Figure 7: We evaluate the generalization ability of other bridge models on other scene images. All models are trained on facial data for image inpainting.

$j$ reduces the impact of degradation. In contrast, tasks with minor degradation (*e.g.*, dehazing) can use lower $j$ to achieve a more stable inference process. In the final stage, lower $j$ values are preferred, as $\hat{x}_0$ significantly influences $x_{i-1}^\circ$, making the final output more sensitive to prediction errors.

Appendix D provides detailed experimental results on timestep configurations for various image restoration tasks. We must acknowledge that the timestep scheduling shown in the Figure 4 is not the optimal choice, but the results presented in Table 1 still demonstrate satisfactory performance.

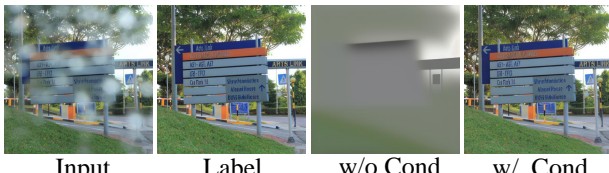

| Input | Label | w/o Cond | w/. Cond |

Figure 8: Ablation of Condition for IRBridge. We visualize the output images before and after applying the condition.

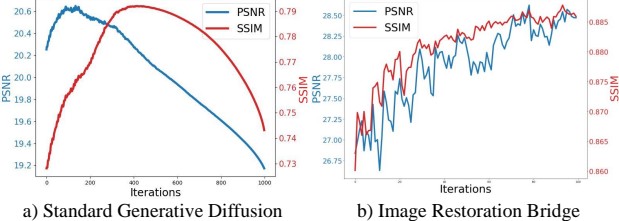

a) Standard Generative Diffusion     b) Image Restoration Bridge

Figure 9: Comparison of the standard generative process starting from noise and the image restoration bridge in terms of PSNR and SSIM.

### 5.2. Discussion about Conditional Guidance

A natural question is whether IRBridge can directly employ unconditional generative models. Figure 8 presents visualization results generated with and without conditional guidance. As shown in the figure, images generated without conditional guidance preserve the overall layout but fail to produce realistic details. The theoretical explanation lies in the transition equation described in Eq 7, which requires a given sample $x_0 \sim P_{data}$ to achieve transitions between states. However, due to the lack of conditional constraints, unconditional models cannot ensure that the estimates $\hat{x}_0$ across all timesteps correspond to the same sample. These inconsistencies among these estimates lead to conflicts during the iterative process, resulting in cumulative errors and causing the final output to converge toward the average of all samples. Visually, as shown in the figure, this produces images that retain only the overall layout but lose nearly all texture and detail. Therefore, introducing conditional guidance in IRBridge is indispensable, as it guides the model in predicting a consistent sample across different timesteps, thereby avoiding the aforementioned issues.

### 5.3. Discussion about Diffusion Process

In Figure 9, we present the PSNR and SSIM curves for samples predicted during the diffusion process using standard generative diffusion (starting from pure Gaussian noise) and the image restoration bridge (starting from degraded images). It can be observed that the curve for standard diffusion first rises and then falls, while the diffusion process with the bridge steadily increases. More importantly, the PSNR/SSIM of the generative diffusion process consistently remains lower than that of the diffusion bridge. Our explanation is that the paradigm starting from pure noise lacks

continuous conditional constraints, making it hard to anchor the degraded image, which causes it to gradually diverge from the target image as the inference progresses. In contrast, the restoration bridge provides continuous conditional constraints, resulting in images that are more consistent with the desired outcome. This result further validates that, compared to generative diffusion starting from pure noise, the bridge trajectory starting from a low-quality image leads to a more effective diffusion process.

### 5.4. Discussion about Reverse Transition

It should be noted that the proposed reverse transition is not the only possible option. After obtaining an estimate for the initial sample $x_0$, the next sample can be obtained through the original reverse process of the image restoration bridge. In general, the sample $x_{i-1}^{\circ}$ obtained by the reverse transition used in IRBridge is not the optimal reverse state that minimizes the negative log-likelihood. Therefore, the iterative path of IRBridge is not the optimal path. However, the results reported in the experiments of this paper are all based on the proposed reverse transition, aiming to provide a more comprehensive understanding of the proposed transition equation.

## 6. Limitations and Future Work

**Limitations.** IRBridge allows for task-specific customization of hyperparameters, but determining optimal values is often a complex and empirically driven process. A recent work(Qiu et al., 2025) propose optimizing diffusion coefficients via local Schrödinger bridges to enhance the performance of pretrained models, which similarly offers a promising path for IRBridge to obtain optimal hyperparameters. However, we argue that empirically chosen coefficients are often sufficient, and conducting complex post-optimization may not be cost-effective.

**Future work.** The ControlNet used in this study is a relatively coarse implementation. We do not optimize the conditional guidance module or introduce task-specific objectives for image restoration. Future work could enhance performance by tailoring the guidance mechanism for IR tasks, as adopted in recent studies (Ai et al., 2024; Rajagopalan et al., 2024). Moreover, further optimization of the VAE can help reduce encoding losses and mitigate the over-smoothing effect caused by the loss of high-frequency details.

We believe that IRBridge is just an early application of the transition equation proposed in Proposition 3.1. With this equation, we can explore directly using the pre-trained generative model as an initialization model for further training with image restoration bridge objectives. We hope this work will contribute to the further application of bridge models in the field of image restoration.

## 7. Related Works

**Diffusion Generative Prior in Image Restoration.** Image restoration methods that integrate diffusion priors typically condition the generation process on degraded images. Some approaches (Zhu et al., 2023b; Feng et al., 2023; Chung et al., 2022; Kawar et al., 2022; Peng et al., 2024) achieve image restoration by solving or approximating $\log \nabla_{x_t} p(y \mid x_t)$, but they often require specific prior knowledge of the degradation process, thereby losing universality. Other efforts (Fei et al., 2023; Gou et al., 2024) aim for unified image restoration by training a degradation model and guiding the process through its gradient, but the introduction of adversarial training leads to instability.

**Bridge Models.** Bridge models construct a stochastic diffusion process between two distributions (Bernton et al., 2019; Liu et al., 2022a; De Bortoli et al., 2021; Liu et al., 2023; Shi et al., 2024; Tang et al., 2024) or fixed endpoints (Li et al., 2023a; Zhou et al., 2023), thereby eliminating the need for prior knowledge about the distributions. Some methods (Delbracio & Milanfar, 2023; Yue et al., 2023b; Luo et al., 2023; Yue et al., 2023a; Liu et al., 2024; Zheng et al., 2024) have proposed applying bridge models to image restoration to learn the transport mapping from degraded images to clear images. However, these approaches still require training a model from scratch for each type of degradation. Our IRBridge primarily addresses this issue.

## 8. Conclusion

This paper proposes a transition equation that bridges two diffusion processes with the same endpoint distribution. By bridging the image restoration bridge with the generative diffusion process, we introduce a novel image restoration framework, IRBridge, which leverages existing pre-trained generative models to solve image restoration bridges. Extensive experiments on various image restoration tasks demonstrate that IRBridge effectively leverages diffusion generative priors and provides more flexible options. We believe that IRBridge will further advance the application of bridge models in image restoration.

## Impact Statement

**Ethical Impacts.** This study does not raise any ethical concerns. The research does not involve subjective assessments or the use of private data. Only publicly available datasets are utilized for experimentation.

**Expected Societal Implications.** This work aims to address the problem of image restoration, primarily by leveraging pretrained image generative models to solve existing image restoration bridges. Since the method relies on pretrained generative models and the proposed transition equation can also be applied to tasks such as image translation, the main ethical concerns focus on the potential misuse of this technology for creating or manipulating deepfakes or inappropriate content. Such misuse may result in the spread of misinformation, violations of privacy, and other harmful outcomes. To reduce these risks, it is essential to establish sound ethical guidelines and maintain continuous oversight. These concerns are not specific to our method but are widely present across many generative tasks.

We suggest that embedding invisible watermarks in the generated images or implementing safety-checking modules can effectively deter misuse and help ensure that proper attribution or confirmation is provided when the images are used.

## Acknowledgements

This work was supported by the National Natural Science Foundation of China (No. 62222211, No.U24A20326).

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

## A. Proof of Proposition 3.1

*Proof.* Given two diffusion processes with same endpoint distribution $P_{data}$, and their forward diffusion processes can be expressed in the reparameterized form:

$$x_i^\circ = f_i^\circ x_0 + b_i^\circ + \sigma_i^\circ \epsilon_1, \tag{8a}$$
$$x_j^* = f_j^* x_0 + b_j^* + \sigma_j^* \epsilon_2, \tag{8b}$$

where $f_i^\circ, f_j^*, b_i^\circ, b_j^*, \sigma_i^\circ, \sigma_j^*$ are known parameters, and $\epsilon \sim \mathcal{N}(0, I)$ is standard Gaussian noise. Considering that their respective diffusion processes are independent of each other, we can write that $x_i^\circ$ and $x_j^*$ are are independent when given a sample $x_0 \sim P_{data}$. That means $p(x_j^* \mid x_0) = p(x_j^* \mid x_i^\circ, x_0)$. We assume the following reparameterization:

$$x_j^* = \alpha \cdot x_i^\circ + \beta \cdot x_0 + \gamma + \sigma \cdot \epsilon, \tag{9}$$

Eq 9 and Eq 8b are essentially equivalent. We substitute Eq 8a into Eq 9, and by using the method of undetermined coefficients, we obtain the following equality:

$$\begin{cases} f_j^* = \alpha \cdot f_i^\circ + \beta, \\ b_j^* = \alpha \cdot b_i^\circ + \gamma, \\ (\sigma_j^*)^2 = (\alpha \cdot \sigma_i^\circ)^2 + \sigma^2, \end{cases} \tag{10}$$

Therefore, we can obtain that:

$$\begin{cases} \alpha = \sqrt{\dfrac{(\sigma_j^*)^2 - \sigma^2}{(\sigma_i^\circ)^2}}, \\ \beta = f_j^* - \sqrt{\dfrac{(\sigma_j^*)^2 - \sigma^2}{(\sigma_i^\circ)^2}} \cdot f_i^\circ, \\ \gamma = b_j^* - \sqrt{\dfrac{(\sigma_j^*)^2 - \sigma^2}{(\sigma_i^\circ)^2}} \cdot b_i^\circ, \end{cases} \tag{11}$$

*Conditions for validity.* We need to emphasize that Eq 7 is still a reparameterization. Therefore, its validity is subject to the following conditions:

$$\begin{cases} 0 \leq \sigma \leq \sigma_j^*, \\ f_j^* - \sqrt{\dfrac{(\sigma_j^*)^2 - \sigma^2}{(\sigma_i^\circ)^2}} \cdot f_i^\circ \geq 0, \end{cases} \tag{12}$$

Through further simplification, we can get that $\max\{0, [(\sigma_j^*)^2 - (\frac{f_j^* \cdot \sigma_i^\circ}{f_i^\circ})^2]\} \leq \sigma^2 \leq (\sigma_j^*)^2$.

At this point, we have completed the proof of Proposition 3.1.

## B. Reparameterization of the Diffusion Coefficients

In this section, we present the reparameterization of both image generative diffusion and image restoration diffusion coefficients, rewriting them in a unified linear combination form as $x_t = f_t x_0 + b_t + \sigma_t \epsilon$. As shown in Table 2, although the objectives of generative and restorative diffusion differ, their defined diffusion paths can both be uniformly expressed as linear Gaussian paths. This creates favorable conditions for applying the transition equation proposed in Eq 7.

It is important to note that, beyond differences in the definitions of diffusion paths, image restoration bridges also differ in their training objectives. For instance, both IRSDE and GOUB adopt the score of their respective reverse SDEs as training targets, which are consistent with those used in generative diffusion models. However, RDDM decouples the residual and noise components, estimating them with separate networks. At this stage, IRBridge does not leverage these differentiated objectives, but instead directly adopts the training objective of generative diffusion to train a conditional generation model. Future work could explore incorporating training objectives specific to restoration bridges to further improve model performance.

| Method | $f_t$ | $b_t$ | $\sigma_t$ |
|---|---|---|---|
| DDPM (Ho et al., 2020) | $\sqrt{\bar{\alpha}_t}$ | $0$ | $\sqrt{1-\bar{\alpha}_t}$ |
| Rectified Flow (Liu et al., 2022b) | $1-t$ | $0$ | $t$ |
| IR-SDE (Luo et al., 2023) | $e^{-\bar{\theta}_t}$ | $(1-e^{-\bar{\theta}_t})\cdot x_{lq}$ | $\lambda^2(1-e^{-2\bar{\theta}_t})$ |
| GOUB (Yue et al., 2023a) | $e^{-\bar{\theta}_t}\frac{\bar{\sigma}_{t:T}^2}{\bar{\sigma}_T^2}$ | $\left[(1-e^{-\bar{\theta}_t})\frac{\bar{\sigma}_{t:T}^2}{\bar{\sigma}_T^2}+e^{-2\bar{\theta}_{t:T}}\frac{\bar{\sigma}_t^2}{\bar{\sigma}_T^2}\right]\cdot x_{lq}$ | $\frac{\bar{\sigma}_t^2\bar{\sigma}_{t:T}^2}{\bar{\sigma}_T^2}$ |
| RDDM (Liu et al., 2024) | $1-\bar{\alpha}_t$ | $\bar{\alpha}_t\cdot x_{lq}$ | $\bar{\beta}_t$ |
| DiffUIR (Zheng et al., 2024) | $1-\bar{\alpha}_t$ | $(\bar{\alpha}_t-\bar{\delta}_t)\cdot x_{lq}$ | $\bar{\beta}_t$ |

Table 2: Reparameterization of different diffusion processes, where $x_{lq}$ denotes the input low-quality image. For detailed definitions of the symbols shown, please refer to the corresponding original paper.

## C. Critical Timsteps across Different Generative Paradigms

In Figure 10, we present the critical timesteps corresponding to representative generative diffusion models, including DDPM(Ho et al., 2020), Stable Diffusion v1-5(Rombach et al., 2022), and Recited Flow(Liu et al., 2022b). Due to cost considerations, we conduct experiments only on Stable Diffusion v1-5. However, the figure shows that the critical timesteps for other models also lie within the typical range covered by pre-trained generative diffusion models. This theoretical consistency supports the general applicability of IRBridge to a broader class of generative diffusion models.

## D. Exploratory Study on Timestep Scheduling Strategies

We first predefined 8 timestep schedules and evaluated the effectiveness of IRBridge on a small batch (batch size of 4) of image samples to determine the optimal timestep selection strategy for different restoration tasks. In Figure 11, we visualize these predefined timestep schedules. The schedules are categorized into three types: **a) One Stage (Setting 1-2):** This strategy employs a single-stage linear decay for the timestep schedule, with two maximum timestep values tested: 850 and 550. **b) Two Stage (Setting 3-5):** This strategy utilizes a two-stage linear timestep schedule, with the primary breakpoints set at 850 and 550. **c) Additive (Setting 6-8):** This strategy adds offsets to the critical timesteps. We tested constant offsets (100 and 300) as well as linearly decreasing timesteps (from 300 to 100).

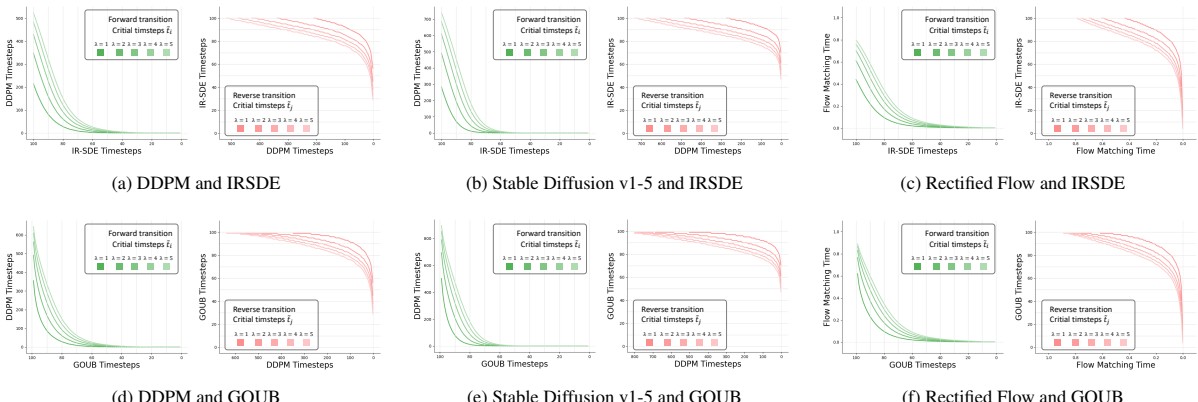

Figure 10: Critical timesteps across different paradigms. We cover commonly used generative models (Stable Diffusion v1-5, DDPM, and Rectified Flow) as well as image restoration bridge models (IR-SDE and GOUB).

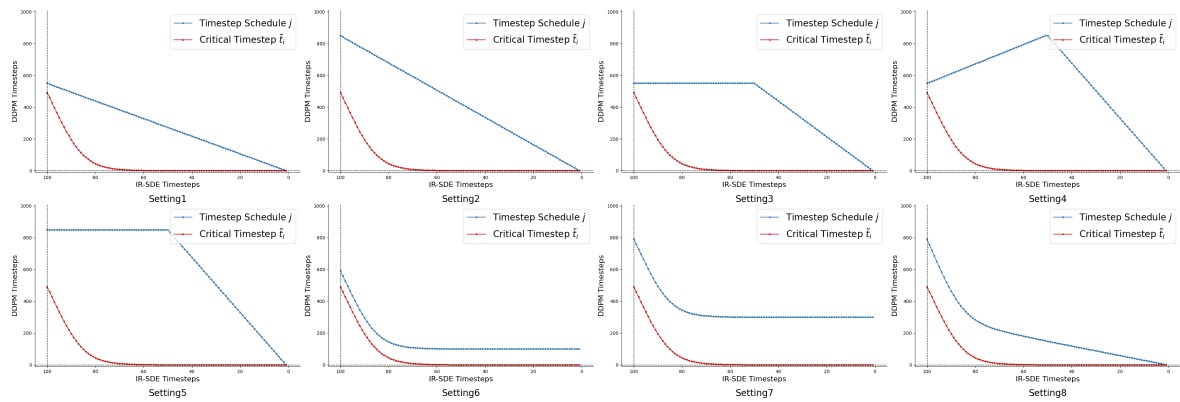

Figure 11: Predefined timestep schedules for exploratory study on timestep scheduling strategies.

We present the results of using 8 predefined timestep schedules across various tasks in Figure (12,13,14,15,16,16,17), including the best and final PSNR and SSIM during inference, as well as their complete evolution curves (for brevity, we only show the curves for IR-SDE). Based on the above observations, we progressively fine-tune the timestep schedules according to the macro-analysis introduced in Section 5.1 to achieve relatively better configurations.

It is important to note that we use small batches (batch size of 4) in these exploratory experiments to save costs, which leads to slight discrepancies between the listed results and the final results shown in Table 1. There are two reasons for this: 1) small-batch samples cannot fully represent the distribution of the entire dataset, and 2) the results listed in Table 1 are obtained with fine-tuned timestep schedules, which are inherently better than the predefined schedules used in the exploratory experiments.

We argue that using small-batch samples in exploratory studies is feasible because our primary focus is on observing the evolution curves of PSNR and SSIM rather than their final values. Therefore, the errors caused by small-batch samples do not significantly affect our conclusions. The timestep schedules we select are typically those that result in steadily increasing metric curves, ensuring a more stable inference process.

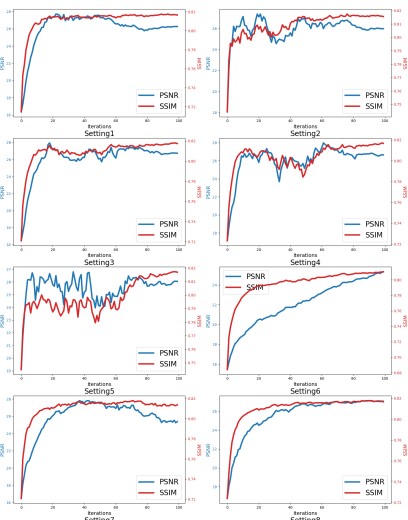

(a) The curves of PSNR and SSIM.

| Type | Setting | Method | last-PSNR | last-SSIM | best-PSNR | best-SSIM |
|---|---|---|---|---|---|---|
| One Stage | Setting1 | IRSDE | 26.2638 | 0.8166 | 27.7116 | 0.8147 |
| | | GOUB | 25.5185 | 0.8156 | 26.9044 | 0.8149 |
| | Setting2 | IRSDE | 25.9933 | 0.8155 | 27.3833 | 0.8081 |
| | | GOUB | 26.0375 | 0.8160 | 27.2455 | 0.8086 |
| Two Stage | Setting3 | IRSDE | 26.7463 | 0.8175 | 27.9550 | 0.8154 |
| | | GOUB | 26.8282 | 0.8175 | 27.4642 | 0.8119 |
| | Setting4 | IRSDE | 26.6423 | 0.8166 | 27.9748 | 0.8121 |
| | | GOUB | 26.8906 | 0.8165 | 27.9497 | 0.8136 |
| | Setting5 | IRSDE | 26.0475 | 0.8168 | 26.8019 | 0.7965 |
| | | GOUB | 25.6752 | 0.8154 | 26.8029 | 0.7965 |
| Additive | Setting6 | IRSDE | 25.3299 | 0.8105 | 25.3299 | 0.8105 |
| | | GOUB | 26.8587 | 0.8177 | 27.1057 | 0.8180 |
| | Setting7 | IRSDE | 25.3820 | 0.8139 | 27.8038 | 0.8166 |
| | | GOUB | 24.9189 | 0.8124 | 27.6537 | 0.8180 |
| | Setting8 | IRSDE | 27.0663 | 0.8167 | 27.1319 | 0.8175 |
| | | GOUB | 26.7459 | 0.8181 | 27.9865 | 0.8188 |

(b) Experimental results of mini-batch samples using predefined timestep scheduling in the image deraining task.

Figure 12: Exploratory study on timestep scheduling strategy for the image deraining task.

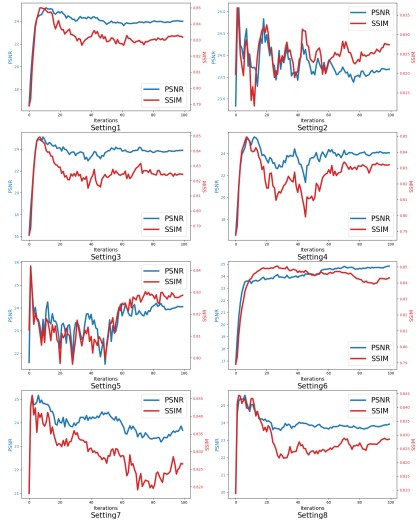

(a) The curves of PSNR and SSIM.

| Type | Setting | Method | last-PSNR | last-SSIM | best-PSNR | best-SSIM |
|------|---------|--------|-----------|-----------|-----------|-----------|
| One Stage | Setting1 | IRSDE | 23.9948 | 0.8317 | 25.1858 | 0.8482 |
| | | GOUB | 23.9132 | 0.8300 | 23.9460 | 0.8305 |
| | Setting2 | IRSDE | 23.6813 | 0.8274 | 25.0897 | 0.8328 |
| | | GOUB | 23.7265 | 0.8277 | 24.8072 | 0.8322 |
| Two Stage | Setting3 | IRSDE | 23.8970 | 0.8243 | 25.1280 | 0.8458 |
| | | GOUB | 23.9113 | 0.8252 | 24.1815 | 0.8266 |
| | Setting4 | IRSDE | 24.0643 | 0.8321 | 25.5087 | 0.8407 |
| | | GOUB | 24.0983 | 0.8340 | 24.4485 | 0.8320 |
| | Setting5 | IRSDE | 24.0563 | 0.8286 | 25.8331 | 0.8419 |
| | | GOUB | 24.0445 | 0.8304 | 24.2596 | 0.8166 |
| Additive | Setting6 | IRSDE | 24.8325 | 0.8431 | 24.8327 | 0.8427 |
| | | GOUB | 22.5133 | 0.8273 | 22.5676 | 0.8244 |
| | Setting7 | IRSDE | 23.6825 | 0.8265 | 25.1649 | 0.8410 |
| | | GOUB | 23.5327 | 0.8255 | 24.2897 | 0.8267 |
| | Setting8 | IRSDE | 23.9262 | 0.8285 | 25.5893 | 0.8432 |
| | | GOUB | 23.8193 | 0.8252 | 24.3383 | 0.8339 |

(b) Experimental results of mini-batch samples using predefined timestep scheduling in the image dehazing task.

Figure 13: Exploratory study on timestep scheduling strategy for the image dehazing task.

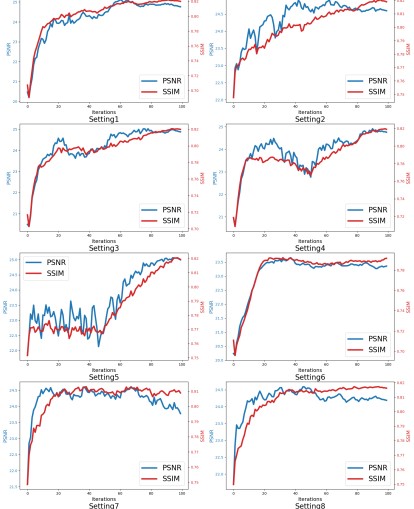

(a) The curves of PSNR and SSIM.

| Type | Setting | Method | last-PSNR | last-SSIM | best-PSNR | best-SSIM |
|------|---------|--------|-----------|-----------|-----------|-----------|
| One Stage | Setting1 | IRSDE | 24.7619 | 0.8200 | 25.1127 | 0.8163 |
| | | GOUB | 24.8466 | 0.8205 | 25.1788 | 0.8167 |
| | Setting2 | IRSDE | 24.5954 | 0.8189 | 24.9139 | 0.8113 |
| | | GOUB | 24.5977 | 0.8192 | 24.9417 | 0.8026 |
| Two Stage | Setting3 | IRSDE | 24.8803 | 0.8198 | 25.0266 | 0.8143 |
| | | GOUB | 24.8851 | 0.8196 | 25.0398 | 0.8158 |
| | Setting4 | IRSDE | 24.7594 | 0.8189 | 24.9132 | 0.8179 |
| | | GOUB | 24.6576 | 0.8177 | 24.7896 | 0.8194 |
| | Setting5 | IRSDE | 24.9924 | 0.8193 | 25.0615 | 0.8193 |
| | | GOUB | 24.9203 | 0.8183 | 25.0298 | 0.8188 |
| Additive | Setting6 | IRSDE | 23.3574 | 0.7916 | 23.6550 | 0.7918 |
| | | GOUB | 24.0109 | 0.7985 | 24.9109 | 0.8078 |
| | Setting7 | IRSDE | 23.7726 | 0.8089 | 24.6086 | 0.8124 |
| | | GOUB | 24.1235 | 0.8097 | 25.1149 | 0.8117 |
| | Setting8 | IRSDE | 24.1872 | 0.8162 | 24.5922 | 0.8137 |
| | | GOUB | 24.3540 | 0.8152 | 24.8247 | 0.8149 |

(b) Experimental results of mini-batch samples using predefined timestep scheduling in the image desnowing task.

Figure 14: Exploratory study on timestep scheduling strategy for the image desnowing task.

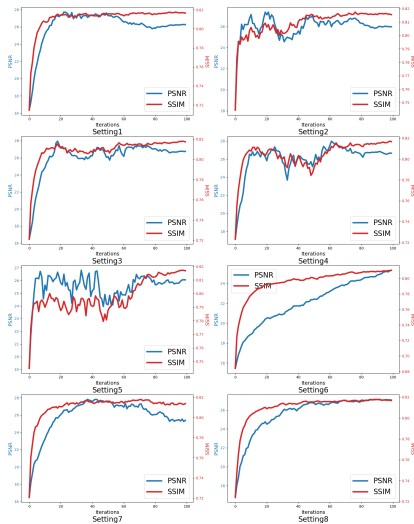

(a) The curves of PSNR and SSIM.

| Type | Setting | Method | last-PSNR | last-SSIM | best-PSNR | best-SSIM |
|------|---------|--------|-----------|-----------|-----------|-----------|
| One Stage | Setting1 | IRSDE | 26.0548 | 0.7923 | 26.8779 | 0.7784 |
| | | GOUB | 25.9791 | 0.7917 | 26.6805 | 0.7789 |
| | Setting2 | IRSDE | 26.2883 | 0.7878 | 26.6549 | 0.7821 |
| | | GOUB | 26.3697 | 0.7892 | 26.6449 | 0.7826 |
| Two Stage | Setting3 | IRSDE | 26.5094 | 0.7884 | 26.7336 | 0.7837 |
| | | GOUB | 26.5697 | 0.7887 | 26.7599 | 0.7849 |
| | Setting4 | IRSDE | 25.7990 | 0.7876 | 26.0884 | 0.7713 |
| | | GOUB | 26.1403 | 0.7876 | 26.2381 | 0.7854 |
| | Setting5 | IRSDE | 25.1104 | 0.7800 | 25.1705 | 0.7802 |
| | | GOUB | 24.8601 | 0.7762 | 24.8668 | 0.7572 |
| Additive | Setting6 | IRSDE | 26.7129 | 0.7935 | 26.7129 | 0.7935 |
| | | GOUB | 25.4206 | 0.7916 | 25.6417 | 0.7913 |
| | Setting7 | IRSDE | 25.9174 | 0.7783 | 27.2205 | 0.7852 |
| | | GOUB | 25.6085 | 0.7790 | 26.8346 | 0.7792 |
| | Setting8 | IRSDE | 27.0162 | 0.7966 | 27.1633 | 0.7928 |
| | | GOUB | 26.5046 | 0.7965 | 26.6997 | 0.7958 |

(b) Experimental results of mini-batch samples using predefined timestep scheduling in the image raindrop removal task.

Figure 15: Exploratory study on timestep scheduling strategy for the image raindrop removal task.

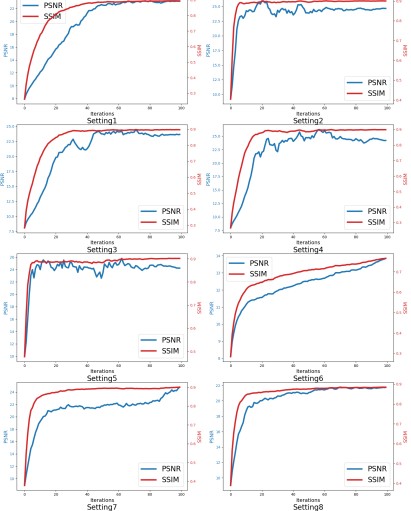
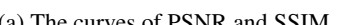

(a) The curves of PSNR and SSIM.

| Type | Setting | Method | last-PSNR | last-SSIM | best-PSNR | best-SSIM |
|------|---------|--------|-----------|-----------|-----------|-----------|
| One Stage | Setting1 | IRSDE | 23.1250 | 0.8931 | 23.1894 | 0.8915 |
| | | GOUB | 25.5623 | 0.9031 | 25.8327 | 0.9027 |
| | Setting2 | IRSDE | 24.6687 | 0.8999 | 25.9736 | 0.8989 |
| | | GOUB | 24.6308 | 0.9003 | 26.0017 | 0.8964 |
| Two Stage | Setting3 | IRSDE | 23.6494 | 0.8971 | 24.4669 | 0.8978 |
| | | GOUB | 23.9677 | 0.8975 | 25.2930 | 0.9003 |
| | Setting4 | IRSDE | 24.1875 | 0.8990 | 26.2098 | 0.8948 |
| | | GOUB | 24.6070 | 0.8995 | 26.1871 | 0.8970 |
| | Setting5 | IRSDE | 24.2217 | 0.9015 | 25.7966 | 0.8990 |
| | | GOUB | 24.4615 | 0.9021 | 25.6628 | 0.8864 |
| Additive | Setting6 | IRSDE | 13.8463 | 0.7655 | 13.8463 | 0.7655 |
| | | GOUB | 24.5219 | 0.9018 | 24.6309 | 0.9010 |
| | Setting7 | IRSDE | 24.7898 | 0.9007 | 24.7898 | 0.9007 |
| | | GOUB | 25.1895 | 0.8986 | 25.2851 | 0.8977 |
| | Setting8 | IRSDE | 21.7411 | 0.8845 | 21.7752 | 0.8825 |
| | | GOUB | 24.4582 | 0.9001 | 25.7583 | 0.8974 |

(b) Experimental results of mini-batch samples using predefined timestep scheduling in the low light enhancement task.

Figure 16: Exploratory study on timestep scheduling strategy for the low light enhancement task.

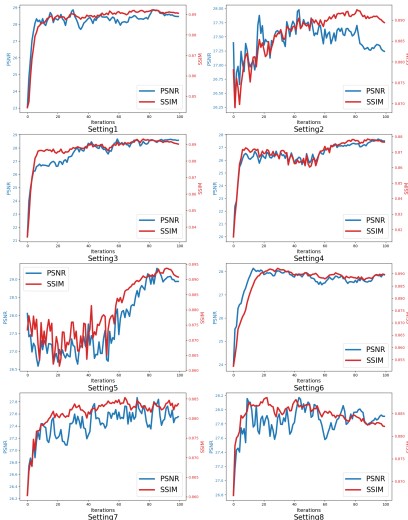

(a) The curves of PSNR and SSIM.

| Type | Setting | Method | last-PSNR | last-SSIM | best-PSNR | best-SSIM |
|------|---------|--------|-----------|-----------|-----------|-----------|
| One Stage | Setting1 | IRSDE | 28.4661 | 0.8906 | 28.8653 | 0.8920 |
| | | GOUB | 28.5011 | 0.8904 | 28.9114 | 0.8903 |
| | Setting2 | IRSDE | 27.2414 | 0.8894 | 27.9769 | 0.8905 |
| | | GOUB | 27.3087 | 0.8899 | 27.9763 | 0.8907 |
| Two Stage | Setting3 | IRSDE | 28.5806 | 0.8902 | 28.6749 | 0.8914 |
| | | GOUB | 28.6183 | 0.8910 | 28.6480 | 0.8915 |
| | Setting4 | IRSDE | 27.5063 | 0.8765 | 27.6593 | 0.8773 |
| | | GOUB | 27.4750 | 0.8775 | 27.4750 | 0.8775 |
| | Setting5 | IRSDE | 28.9435 | 0.8908 | 29.3078 | 0.8922 |
| | | GOUB | 28.9642 | 0.8924 | 29.2503 | 0.8923 |
| Additive | Setting6 | IRSDE | 27.8617 | 0.8897 | 28.1424 | 0.8855 |
| | | GOUB | 28.1664 | 0.8911 | 28.6311 | 0.8931 |
| | Setting7 | IRSDE | 27.5545 | 0.8837 | 27.8692 | 0.8846 |
| | | GOUB | 27.7256 | 0.8840 | 27.8662 | 0.8846 |
| | Setting8 | IRSDE | 27.9057 | 0.8822 | 28.1693 | 0.8865 |
| | | GOUB | 28.6027 | 0.8840 | 29.3940 | 0.8905 |

(b) Experimental results of mini-batch samples using predefined timestep scheduling in the image inpainting task.

Figure 17: Exploratory study on timestep scheduling strategy for the image inpainting task.

## E. Experimental details

### E.1. Training Details.

As introduced in Section 4, we train ControlNet to integrate conditional guidance into the pre-trained Stable Diffusion model. We follow the standard training paradigm of ControlNet, fitting the noise added during the forward process. The same settings are applied across all tasks. We use the AdamW optimizer with a learning rate of $5.0 \times 10^{-5}$ and train for a total of 10k steps. The Adam optimizer is used with $\beta_1 = 0.9$ and $\beta_2 = 0.999$ to maintain a balance between the momentum term and the variance estimate. A weight decay of $1.0 \times 10^{-2}$ is applied for regularization, while a small epsilon value of $1.0 \times 10^{-8}$ ensures numerical stability.

The model is trained on an Nvidia RTX 3090 GPU with a batch size of 12. To reduce GPU memory usage during training, mixed-precision training is employed. To manage the learning rate, a constant schedule is employed with 500 warmup steps to gradually ramp up the learning rate at the start of training. The scheduler includes a single cycle with a polynomial decay power of 1 to allow steady updates.

### E.2. Datasets.

We provide details of the datasets used in this section. During both training and testing, all image samples are resized and cropped to a resolution of 512 to match the input requirements of Stable Diffusion. Although we adopt a unified resolution setting, our method can also support higher-resolution inputs, thanks to a patchify strategy similar to that used in WeatherDiff (Özdenizci & Legenstein, 2023).

**Image Deraining.** The image deraining task aims to restore images degraded by rain streaks and similar effects. Due to the difficulty of obtaining real-world paired datasets, most image deraining datasets are synthetically generated. For this task, we used the Rain100H dataset (Yang et al., 2019), which provides 1,800 paired images for training and 100 for testing. The degradations in Rain100H are artificially synthesized as light, thin lines that simulate natural rain streaks.

**Image Dehazing.** To evaluate IRBridge on image dehazing, we used the RESIDE dataset (Li et al., 2019). The model was trained on the Outdoor Training Set (OTS) subset, containing 72,135 images, and tested on the Synthetic Objective Testing Set (SOTS) subset, which includes 500 images. These subsets feature synthetically generated hazy images designed to mimic real-world atmospheric conditions.

**Image Desnowing.** For the image desnowing task, we used the Snow100K dataset (Liu et al., 2018), which includes 100k synthesized snowy images paired with corresponding snow-free ground truth images. Specifically, 50K images are used for training and 50K for testing. The degradations in this dataset simulate snow particles and atmospheric scattering effects,

providing diverse and challenging scenarios for evaluation.

**Image Raindrop Removal.** We evaluated image raindrop removal using the RainDrop dataset (Qian et al., 2018), which comprises 861 training images and 58 testing samples from its Testing Set A subset. The degradations in this dataset simulate the refraction and distortion effects caused by raindrops of varying sizes on camera lenses, presenting unique challenges compared to other weather-related tasks.

**Low light enhancement.** The low-light enhancement task was evaluated using the LOL dataset (Wei et al., 2018). This dataset includes 485 paired images for training and 15 for testing, with each pair consisting of low-light and normal-light versions of the same scene. The images contain noise from low-light capture conditions, with significantly lower mean pixel values than normal images. These characteristics pose challenges for generation models, which must accurately enhance brightness while preserving details.

**Image Inpainting.** Image inpainting experiments were conducted on the CelebA-HQ (Liu et al., 2015) 256×256 dataset. We trained the model using 20,000 images with randomly generated brush masks. This task is particularly challenging due to the random nature of the masks, which disrupt the original image layout and cause significant shifts in mean and variance. The task evaluates the model's ability to generate plausible content while maintaining consistency with the surrounding image context.

## F. Inference Efficiency

Since IRBridge leverages pretrained diffusion models and adopts an iterative diffusion-style inference process, we acknowledge that its inference efficiency is not a primary advantage. Nevertheless, IRBridge provides considerable flexibility by decoupling model training from inference, allowing independent configuration of diffusion coefficients and the number of inference steps. In Table 3, we present the inference time and quantitative results of IRBridge under different step settings. It is worth noting that even with a 75% reduction in inference steps, IRBridge still maintains comparable performance, demonstrating its potential for more efficient inference.

| Inference Steps | 100 | 50 | 25 |
|---|---|---|---|
| PSNR↑ | 29.63 | 29.59 | 29.24 |
| SSIM↑ | 0.876 | 0.877 | 0.858 |
| LPIPS↓ | 0.173 | 0.179 | 0.178 |
| Inference Time | 14.2s | 7.3s | 3.4s |

Table 3: Quantitative results under different inference step settings.

## G. Comparison on Real-world Degraded Images

We compare IRBridge with other related methods on real-world degraded images, and the results are presented in Table 4. Specifically, we used real-world datasets, including RealRain-1K (Li et al., 2022) for deraining, realistic snowy images from Snow100K (Liu et al., 2018) for desnowing, and RTTS (Li et al., 2019) for dehazing. Since these datasets lack ground-truth references, we adopt widely-used no-reference image quality metrics, including MUSIQ(Ke et al., 2021), BRISQUE (Mittal et al., 2011), and NIQE(Mittal et al., 2012), to evaluate the performance of different methods.

As shown in Table 4, IRBridge consistently outperforms other training-from-scratch approaches in real-world scenarios. Even when compared with the recent state-of-the-art method DCPT, IRBridge achieves superior performance, demonstrating its effectiveness under realistic conditions. We attribute this improvement to the integration of pretrained generative priors, which endow the model with enhanced generalization capabilities.

In Figure (18, 19, 20), we present the visual results of IRBridge on real-world deraining, dehazing, and desnowing tasks, providing an intuitive demonstration of its generalization ability in real-world scenarios.

Table 4: Comparison with other methods on real-world datasets.

| Method | RealRain-1k | | | RealSnow subset of Snow100K | | | RTTS | | |
|---|---|---|---|---|---|---|---|---|---|
| | MUSIQ↑ | BRISQUE↓ | NIQE↓ | MUSIQ↑ | BRISQUE↓ | NIQE↓ | MUSIQ↑ | BRISQUE↓ | NIQE↓ |
| IRSDE(Luo et al., 2023) | 33.187 | 46.882 | 12.551 | 39.854 | 40.872 | 10.874 | 45.256 | 52.825 | 9.852 |
| GOUB(Yue et al., 2023a) | 39.758 | 42.261 | 10.641 | 48.289 | 34.824 | 4.987 | 47.952 | 53.578 | 11.587 |
| RDDM(Liu et al., 2024) | 41.975 | 41.870 | 9.587 | 47.586 | 33.879 | 6.881 | 49.888 | 51.574 | 10.054 |
| DiffUIR(Zheng et al., 2024) | 41.758 | 43.821 | 9.001 | 49.812 | 32.987 | 5.027 | 50.517 | 50.954 | 8.805 |
| DCPT(Hu et al., 2025) | 44.915 | 39.843 | 8.147 | 45.812 | 35.871 | 6.871 | 48.109 | 54.641 | 9.847 |
| IRBridge | 43.439 | 39.915 | 8.393 | 51.969 | 32.347 | 4.002 | 52.076 | 41.315 | 5.003 |

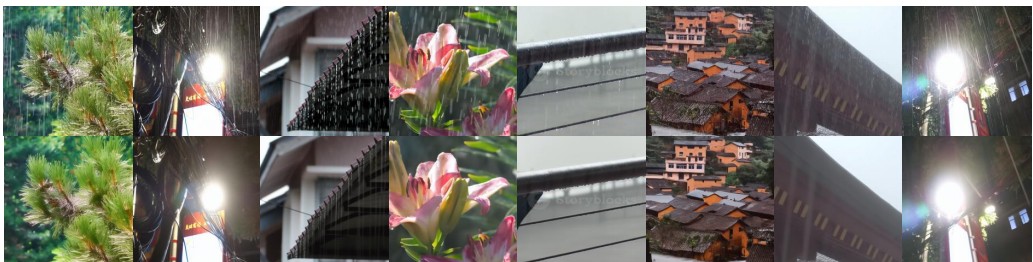

Figure 18: Visualization results of IRBridge in real-world rainy images.

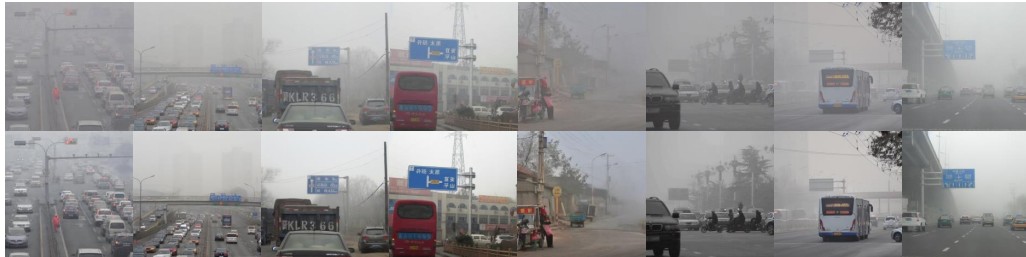

Figure 19: Visualization results of IRBridge in real-world hazy images.

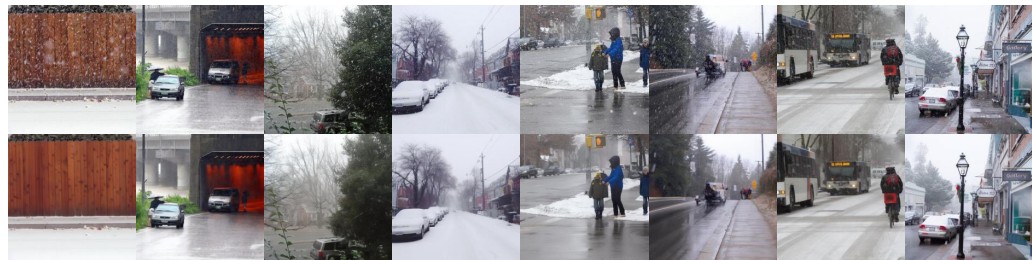

Figure 20: Visualization results of IRBridge in real-world snowy images.

## H. Visualization results for different tasks.

In this section, we present the visualization results of IRBridge on image deraining, dehazing, desnowing, raindrop removal, low-light enhancement, and image inpainting tasks. It is important to note that these results are **unselected** to rigorously evaluate the performance of IRBridge.

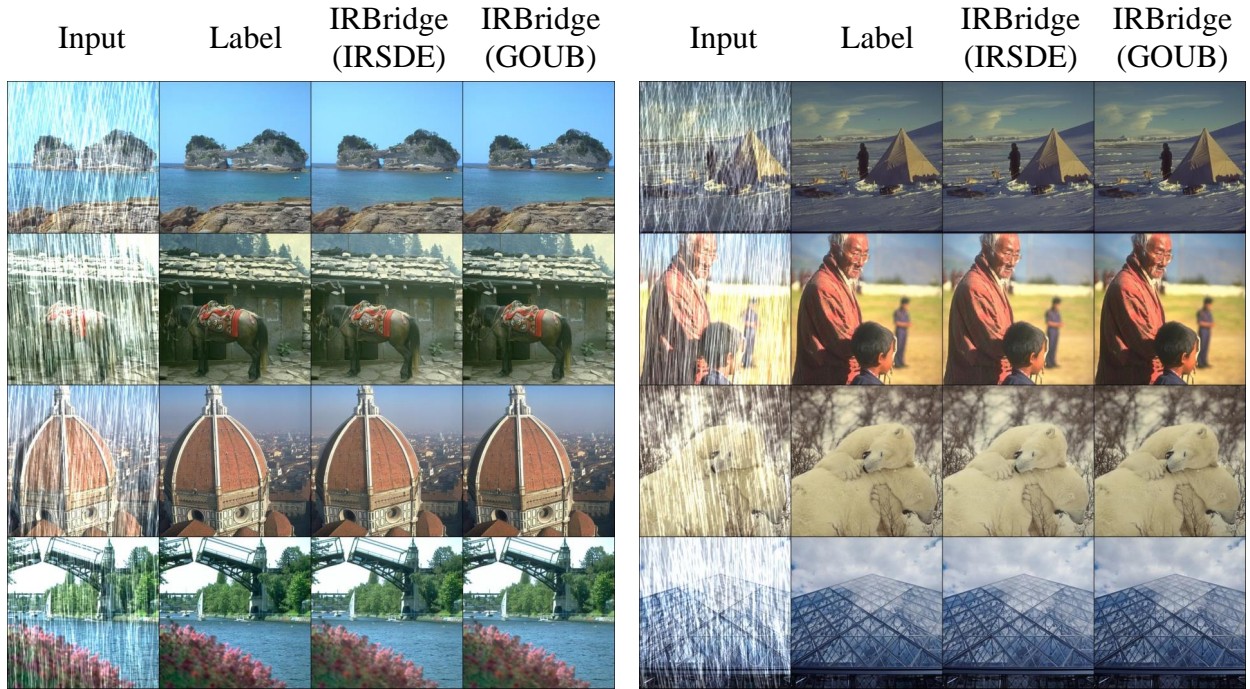

Figure 21: Visualization results of IRBridge in the image deraining task.

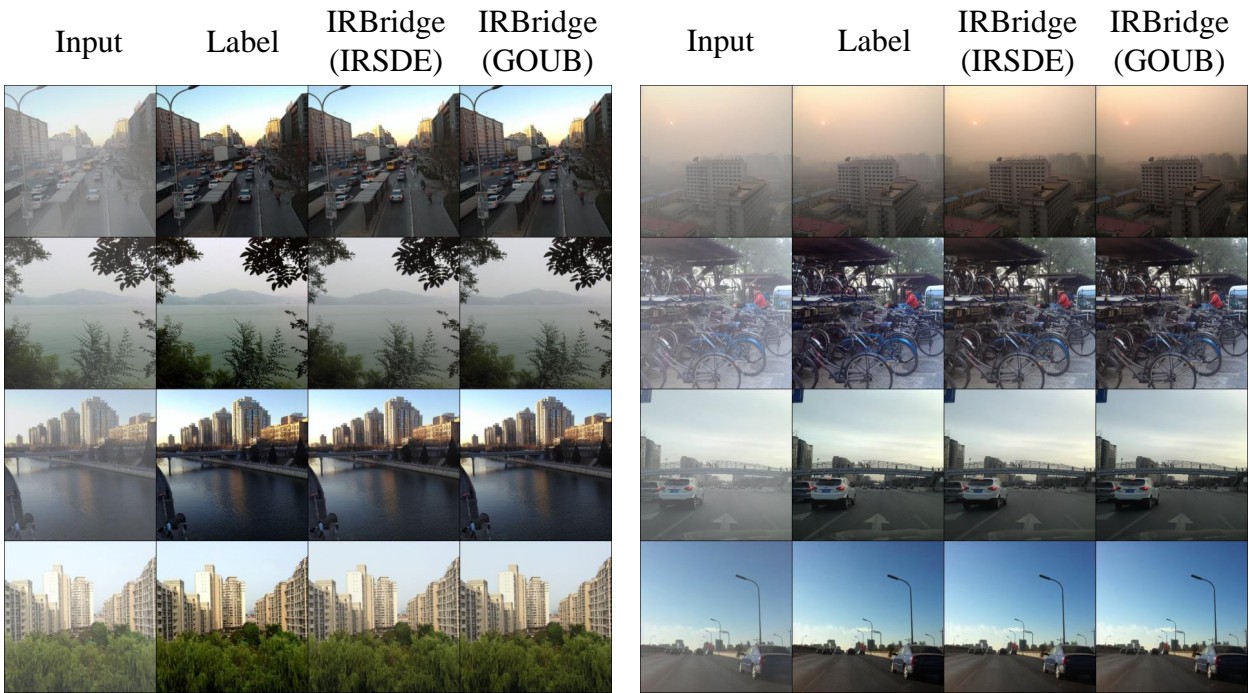

Figure 22: Visualization results of IRBridge in the image dehazing task.

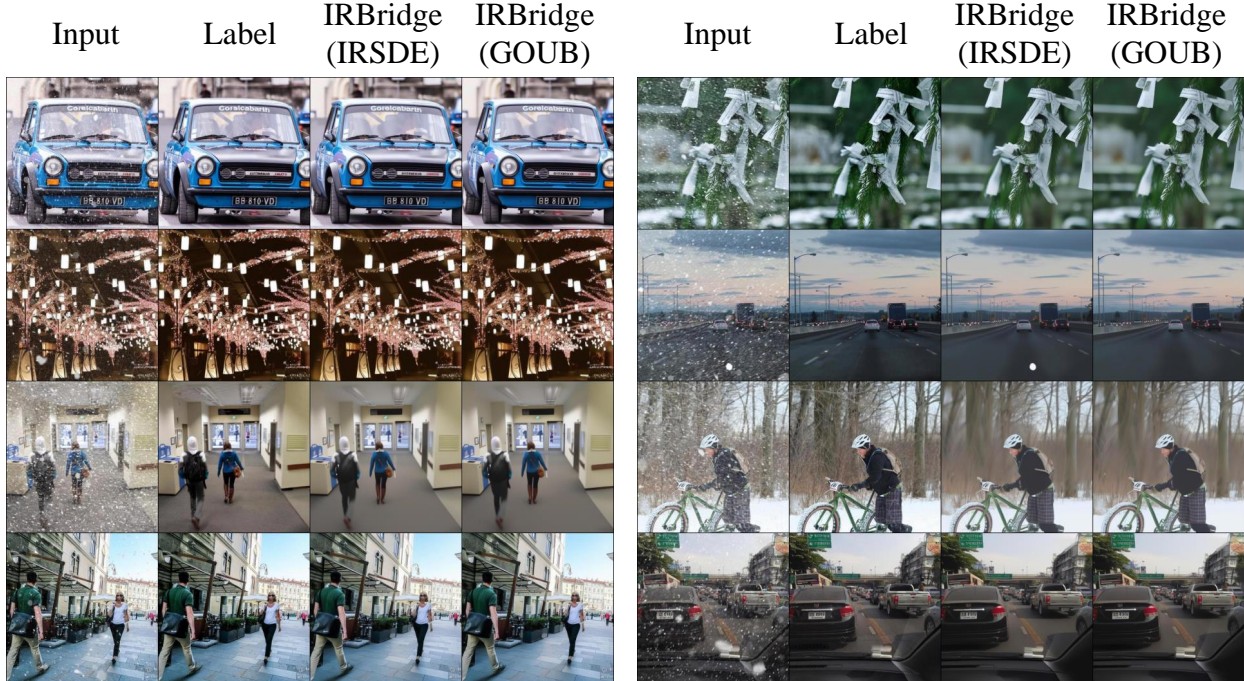

Figure 23: Visualization results of IRBridge in the image desnowing task.

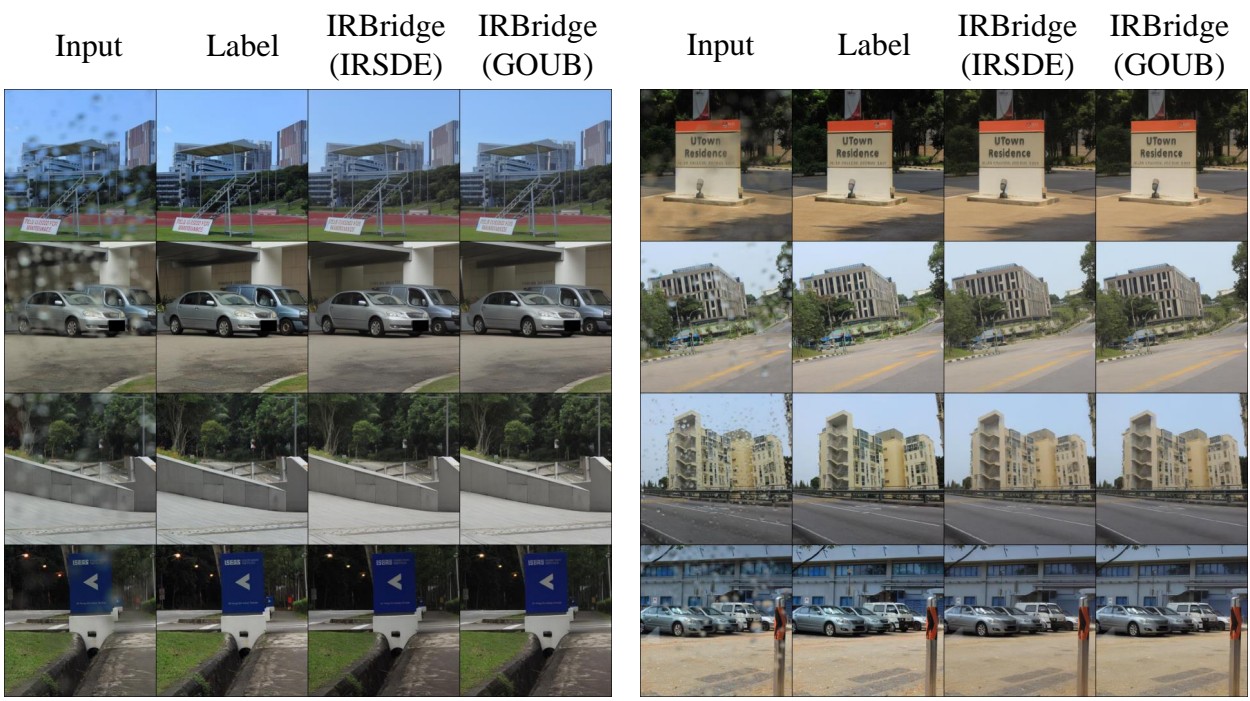

Figure 24: Visualization results of IRBridge in the image raindrop removal task.

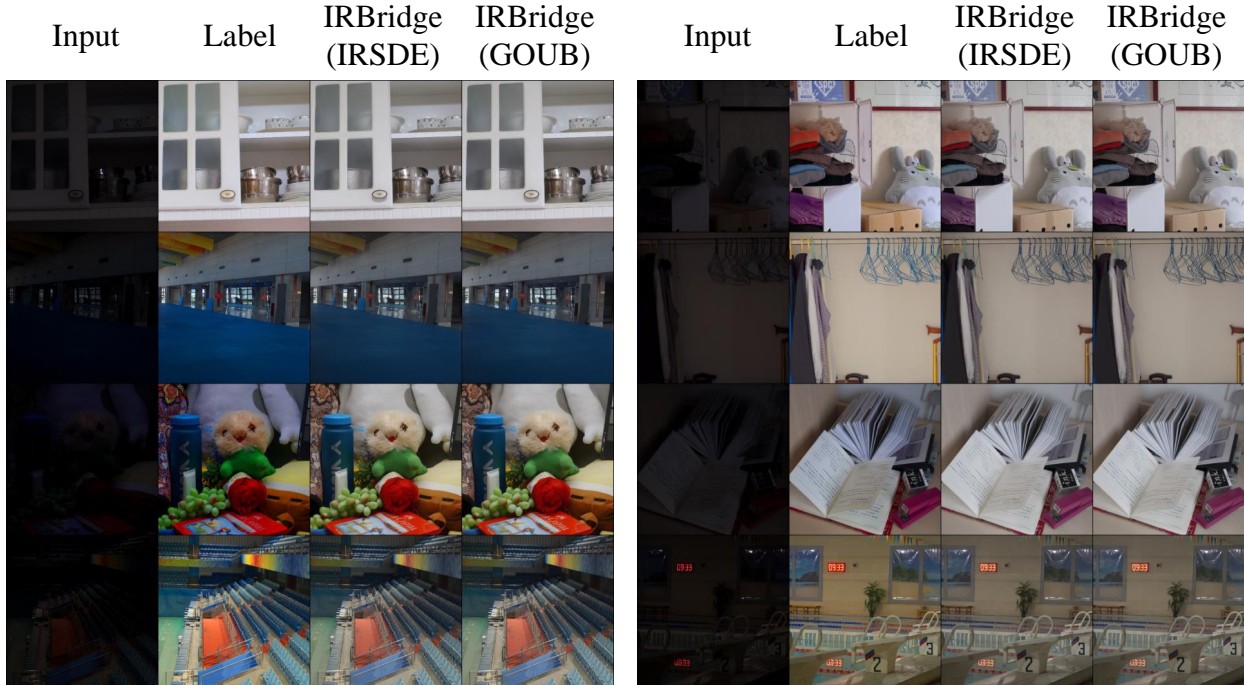

Figure 25: Visualization results of IRBridge in the low light enhancement task.

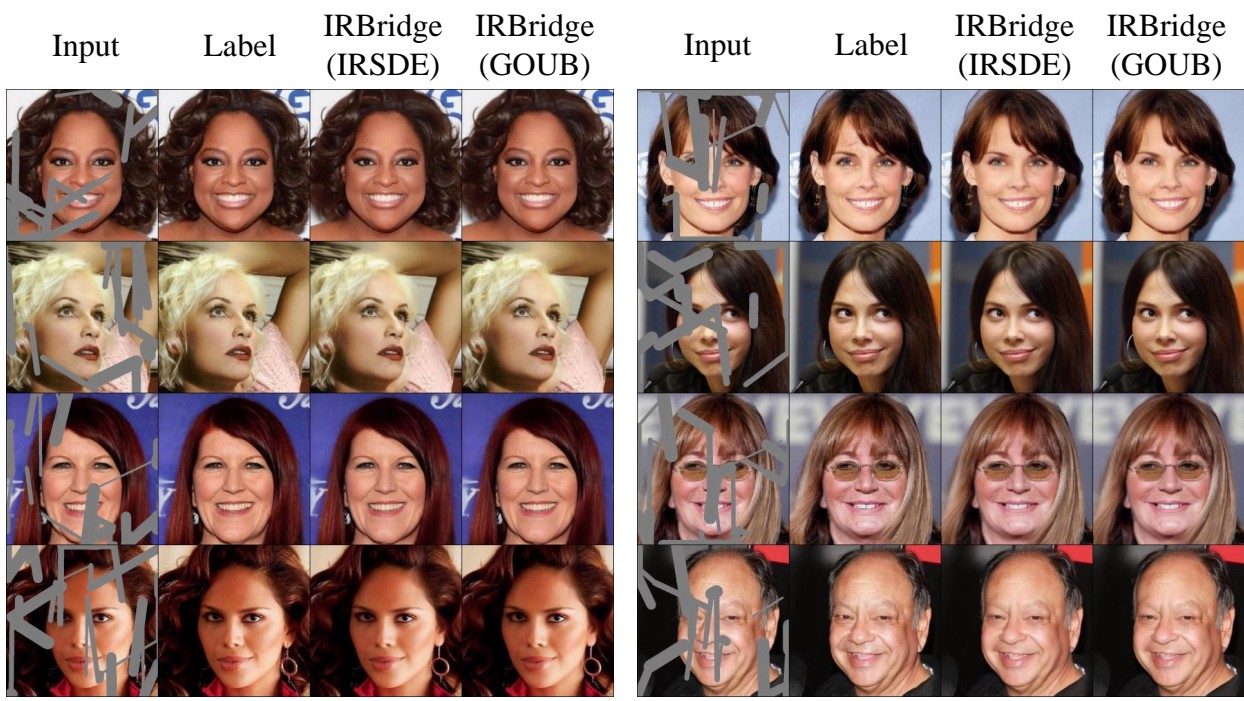

Figure 26: Visualization results of IRBridge in the image inpainting task.

## I. Feasible Improvements and Potential Applications

**Future Work.** As previously mentioned, the current implementation of IRBridge represents only a preliminary attempt at applying the proposed transition equation. Future work can focus on improving the guidance module, adapting VAEs to image restoration tasks, and incorporating additional training objectives tailored for the bridge diffusion process used during inference. These improvements can further enhance the performance of IRBridge on image restoration tasks, and several existing works (Ai et al., 2024; Rajagopalan et al., 2024; Qiu et al., 2025; Lin et al., 2024b; Wang et al., 2025b; Liu et al., 2024; Wang et al., 2025a) may offer valuable insights in these directions. Moreover, the current version of IRBridge does not incorporate textual conditions, leaving the powerful text-guidance capabilities of pretrained diffusion models underutilized. Some existing studies have already demonstrated the feasibility of leveraging additional modalities to improve the performance of image restoration tasks. Building upon prior research (Lin et al., 2023a;b; Huang et al., 2024a; Fu et al., 2024), incorporating multimodal information can further enhance the performance of IRBridge. Additionally, leveraging a broader variety of datasets (Rajagopalan et al., 2024; Huang et al., 2024b; Feng et al., 2024) could also contribute significantly to its improvement.

**Potential Applications.** We highlight that the proposed transition equation is not limited to bridging restoration diffusion and generative diffusion; it is applicable to a wider range of stochastic diffusion processes, as long as the endpoint distributions are consistent. Therefore, we believe this work can also benefit a broader range of applications, such as image editing(Li et al., 2023b; Ju et al., 2024), image translation(Zhou et al., 2023), visual generation(Lin et al., 2024a) and multimodal generation (Ji et al., 2024).

