# OpenReview forum: "IRBridge: Solving Image Restoration Bridge with Pre-trained Generative Diffusion Models"
_ICML.cc/2025/Conference — ICML 2025 poster_

### Official Review · Reviewer_7WdJ · 2025-02-18

**Overall Recommendation:** 3

**Summary:**

This work presents an image restoration framework IRBridge. IRBridge connects the bridge model with the diffusion process of the generative diffusion model by introducing a transition equation, thereby enabling direct utilization of the pre-trained generative model for image inpainting. This methods transforms the state of the bridge model into that of the generative diffusion model through forward and backward transitions, and then exploits the powerful prior knowledge of the generative model for image inpainting. The experiments in the paper demonstrate that IRBridge performs better in various image restoration tasks and exhibits superior robustness and generalization capabilities.

**Claims And Evidence:**

Yes, this work presents a rather detailed theoretical derivation process of the transition equation and a reasonably well-structured introduction to the framework construction of the IRBridge.

**Essential References Not Discussed:**

No, the paper makes a relatively comprehensive citation of the literature.

**Experimental Designs Or Analyses:**

Yes, I perused the work's elaboration on the setup of the experiment, the implementation method of the comparative experiment, and the analysis of other details of the experiment. Specific issues regarding the experimental section can be found in the "weaknesses" part.

**Methods And Evaluation Criteria:**

Yes, this work presents a relatively rich set of comparative experiments to verify the performance of IRBridge and conducts a certain degree of analysis on the experimental details.

**Other Comments Or Suggestions:**

1. There are some inconsistencies in expression. For instance, the vast majority uses ‘timesteps’, while in some places ‘time steps’ is used. It is recommended to maintain consistency here.
2. In the left-hand image of Figure 5, it is rather difficult to discern the values on the abscissa of the two pictures. The numerical range of the timesteps here is of crucial importance.

**Other Strengths And Weaknesses:**

Strengths:
1. The IRBridge framework is novel. It obviates the need to train models from scratch for each degradation type and directly leverages pre - trained generative diffusion models, remarkably reducing the training cost.
2. Through the proposed transition equation, IRBridge can adapt to diverse image restoration tasks and permits the adjustment of hyperparameters during inference to optimize performance.
3. The paper validates the performance and robustness of IRBridge through reasonable comparative experiments, and its performance surpasses that of off-the-shelf methods. Weeknesses:
1. In this work, the selection of hyperparameters is based on empirical methods rather than systematically determining the optimal choices. This brings certain limitations when dealing with some complex degradations or in practical applications.
2. In Figure 3, the innovative points of the article are not well highlighted within the framework of the IRBridge. For instance, the role played by the transition equation in the IRBridge could be emphasized.

**Questions For Authors:**

1. Although the IRBridge reduces the need to train models from scratch, multiple iterations are still required during the inference process, especially when dealing with high-resolution images. Is there any comparison or analysis regarding the consumption of computational resources?
2. Is there a test on the image restoration effect of the IRBridge in more complex degradation scenarios? For example, the results of tasks such as real-world degradation tasks to further validate the practicality of the model.
3. Has the work considers reducing the dependence on the initial state estimation by introducing additional prior information from different perspectives (such as degradation models or image structural information)? Or can the estimation errors of the initial state be gradually corrected through iterative optimization?
If the weeknesses and questions be adequately resolved, I would be delighted to increase my score.

**Relation To Broader Scientific Literature:**

In response to the task of image restoration, previous research aimed to more intuitively simulate the transition from low-quality images to high-quality ones. The advancements in generative diffusion models, such as various types of diffusion models, have demonstrated remarkable potential in this tasks. The bridge model has attracted attention as it simulates the stochastic process between two distributions or fixed points.

**Theoretical Claims:**

Yes, I perused the introduction of the bridge model in the article and the theoretical derivation process of the transition equation thus elicited. Apart from some minor issues (detailed in suggestions), the argumentation is relatively sound.

---

> ### Author Rebuttal · Authors · 2025-04-01
>
> We are grateful for your positive review! We will carefully address your concerns below.
>
> **Q1: Empirical selection of hyperparameters**
>
> We acknowledge that this is a limitation of IRBridge, and we offer two key insights.
>
> + As shown in Appendix C, while different choices of timesteps can significantly affect the intermediate states of the diffusion trajectory, their impact on final results is generally limited.  This suggests that although we may not adopt theoretically optimal inference hyperparameters, the actual performance loss is often acceptable (~5%). Based on our experience, the hyperparameters used for the inpainting task are effective for most tasks, and we refer it as a default setting.
> + Recent works have provided ways to systematically optimize hyperparameters under certain metrics. For example, LDSB [1] recently proposed using local Schrödinger bridge to optimize the diffusion coefficients. IRBridge can benefit from it to obtain optimal hyperparameters in the context of relative entropy minimization.
>
> While optimal hyperparameters are attainable, we believe that the cost of complex optimization is not cost-effective, as manually designed parameters are typically sufficient.
>
> **Q2: Inference efficiency and computational resources.**
>
> In our implementation, we use the SD1.5 model, which processes approximately 7.14 steps per second on a 512×512 image using an RTX 3090 GPU, consuming around 6300MB of VRAM. In [THIS TABLE](https://anonymous.4open.science/r/IRBridge-4181/asserts/table3.md) , we present the performance and inference time of IRBridge under different inference steps. Encouragingly, even with a 75% reduction in inference steps, IRBridge shows minimal performance drop (less than 3%) while significantly reducing inference time (14.2s → 3.4s), demonstrating its potential for improving inference efficiency.  Future work can also focus on compression of the pretrained model to further enhance the inference efficiency.
>
> **Q3: Generalization for real-word degradation.**
>
> We present the evaluation results on real-world datasets in [THIS TABLE](https://anonymous.4open.science/r/IRBridge-4181/asserts/table2.md). We utilized no-reference image quality metrics to evaluate the performance of the methods. As shown in the table, IRBridge demonstrates a clear advantage over other training-from-scratch methods in real-world scenarios. Compared to the latest method, DCPT (ICLR 2025), IRBridge still achieves a better performance (with average 5.6% performance improved), highlighting its superior performance under real-world conditions. We attribute these improvements to the pretrained generative prior that equips the model with stronger generalization capability. Visual examples are provided [HERE](https://anonymous.4open.science/r/IRBridge-4181/asserts/real/show.md).
>
> **Q4: Has the work considers reducing the dependence on the initial state estimation by introducing additional prior information from different perspectives?**
>
> We would like to clarify that, mathematically the dependence on the initial state is solely determined by β in Eq.7. In principle, we only consider the case where β  takes its minimum value, as this theoretically implies minimal reliance on x0.
>
> However, we are open to the idea of retaining some prior information to reduce dependence on x0. For example, given a known degradation model, a plug-and-play (PnP) [2] style approach can be adopted to enforce observation consistency at each iteration step, thereby reducing the degrees of freedom in estimation. Similarly, in the case of image inpainting, we can follow the strategy of RePaint [3] by directly replacing the corresponding pixels in the generated x0 with the uncorrupted pixels from the original LQ sample, thus lowering the reliance on the model’s estimate of initial state.
>
> **Q5: Can the estimation errors of the initial state be gradually corrected through iterative optimization?**
>
> Diffusion-based image generation and restoration methods can generally be viewed as progressively refining their estimate of the initial state during the reverse iteration process. IRBridge, by nature, leverages a pretrained generative model to simulate the reverse process of a restoration bridge. As such, its inference procedure inherently serves to iteratively correct the estimation error of the initial state.
>
> **Q6: Inconsistences in expression and the quality of images and tables.**
>
> We sincerely appreciate the reviewer’s thoughtful suggestions! We will address these issues in the revised version. These constructive suggestions are invaluable for enhancing the overall quality of our work.
>
> [1] Finding Local Diffusion Schrodinger Bridge using Kolmogorov-Arnold Network, CVPR 2025
>
> [2] Denoising Diffusion Models for Plug-and-Play Image Restoration, CVPR2023
>
> [3] RePaint: Inpainting using Denoising Diffusion Probabilistic Models, CVPR 2022

---

> > ### Comment · Reviewer_7WdJ · 2025-04-05
> >
> > The reviewer sincerely thanks the authors for their rebuttal.
> >
> > After carefully reading the rebuttal, most of my concerns have been solved. However, when I check the show.md, there are lots of blurred points in the output image. Please discuss the potential reasons. Moreover, failure cases should also be showed and discussed.

---

> > > ### Author Response · Authors · 2025-04-05
> > >
> > > We appreciate your insightful feedback! Next, we will address your concerns point by point.
> > >
> > > **Q1: Blurred points in the output image**
> > >
> > > **A1.** We acknowledge that some output images do exhibit blurriness, and we offer two explanations for this:
> > >
> > > + **1) The encoding-decoding loss of the VAE.** The Stable Diffusion model we use (similar to most large-scale pretrained generative models) relies on a VAE to compress images to alleviate the computational load during pretraining. However, the VAE inevitably loses some high-frequency details.  While we exclude the errors introduced by the VAE when computing quantitative metrics, these losses are still reflected in the visualized results. To address this issue, we suggest that the VAE can be fine-tuned to mitigate the errors. For example, [1] proposed a Detail Refinement Module (DRM) to enable direct encoder-to-decoder information transformation through a skip connection. Similarly, [2] introduced a Structure Correction Module (SCM) to reverse VAE-induced distortions. We believe that similar approaches could be adopted to correct the errors introduced by the VAE.
> > >
> > >   To validate our claim, we performed a simple fine-tuning of the VAE and presented the visual results in [Fig. 1](https://anonymous.4open.science/r/IRBridge-4181/asserts/reply.md). We simplified the aforementioned approaches by concatenating encoder features of the LQ image to the corresponding decoder layers, and used a zero-initialized convolutional layer to project the output. As shown, even this simple fine-tuning strategy significantly improves the fidelity of fine details in the output images. In future work, we plan to explore more effective ways to enhance the VAE and make it better suited for IR tasks.
> > >
> > > + **2) The impact of the training data.** The visual results we present are generated by models trained on synthetic datasets. These datasets use hand-crafted degradation models to simulate image degradation, which still exhibit a non-negligible gap from real-world degradations. Although IRBridge demonstrates better generalization to real-world scenarios compared to other methods, we acknowledge that the influence of training data still persists in IRBridge. Some subtle degradations may not be completely removed.
> > >
> > > **Q2: Failure cases should also be showed and discussed.**
> > >
> > > Thanks for pointing that out! We acknowledge that IRBridge has some failure cases. As shown in [Fig.2 and Fig.3](https://anonymous.4open.science/r/IRBridge-4181/asserts/reply.md), we present two extreme failure cases.
> > >
> > > + **1) Inconsistent color tones.** Due to the generative nature of our model and the stochastic inference process (induced by noise added during the forward transition),  it tends to produce diverse outputs. In extreme cases, this may result in color tones that are significantly inconsistent with the target image. While this does not affect the visual quality of the output or the performance on downstream tasks (such as object detection), it does lead to a drop in fidelity. Given that our current model is trained purely with the generative objective of DDPM and does not incorporate any restoration-specific priors, we believe that explicitly injecting degradation-related priors could help constrain the model’s degrees of freedom and reduce such inconsistencies (as mentioned in the **response to Q4 in the initial review**).
> > > + **2) Performance bottleneck caused by the quality of training data.** In certain datasets (such as the ITS subset of RESIDE), the ground truth images themselves contain degradations, which can mislead the model during training. In our case, since the target images were captured under foggy conditions, IRBridge inherits the imperfect priors present in the original dataset, making it unable to completely remove the degradations.
> > >
> > > **Fortunately, the two failure cases mentioned above are not common.** Among all the samples in the low-light enhancement task, only the example shown exhibits a noticeable tonal discrepancy, the rest of the output samples are all consistent with the ground truth. In the revised version, we will add a new section in the appendix to discuss the failure cases and explore potential solutions. We sincerely appreciate your constructive feedback!
> > >
> > > If you have any further concerns, please don’t hesitate to reach out — we will do our utmost to respond as promptly as possible.
> > >
> > > **Once again, we truly appreciate your positive and constructive feedback! Your suggestions have been tremendously helpful in improving the quality of our manuscript and have also deepened our understanding of the proposed work.**
> > >
> > >
> > >
> > > [1] Multimodal Prompt Perceiver: Empower Adaptiveness, Generalizability and Fidelity for All-in-One Image Restoration, CVPR 2024
> > >
> > > [2] GenDeg: Diffusion-Based Degradation Synthesis for Generalizable All-in-One Image Restoration, CVPR 2025

---

### Official Review · Reviewer_1Y3v · 2025-03-09

**Overall Recommendation:** 3

**Summary:**

This paper argues that the existing diffusion-based restoration methods are based on the standard diffusion process and cannot intuitively simulate the transition from low-quality images to high-quality images. To solve this problem, the bridge model is used. Subsequently, the paper extends the bridge model to the all-in-one image restoration task and conducts experiments.

**Claims And Evidence:**

This paper highlights that, it:

1. connects the diffusion model and bridge model in the context of image restoration.
2. trains an all-in-one image restoration bridge.
3. demonstrates its effectiveness.

I agree that the first point is the contribution and highlight of this paper, but I disagree with the last two points. The reasons are as follows:

- This paper does not clearly show what difficulties must be solved in training an all-in-one image restoration bridge. A model concurrently trained on multiple datasets can achieve somewhat good restoration performance. I don't think `training an all-in-one image restoration model` can be regarded as a contribution.
- This paper only compared its method with very few methods. The experiments are seriously insufficient.

**Essential References Not Discussed:**

N/A

**Experimental Designs Or Analyses:**

The ablation experiments in this paper are extremely comprehensive, but the comparison with other methods in the main experiment is seriously insufficient.

For example,

- RDDM [1] and DiffUIR [2]. They modified the forward and reverse formulas of the Diffusion Model and were able to transition from low quality images to high quality images. This paper should compare with them.
- DCPT [3]. As a concurrent work, the performance of this paper should be on par with DCPT. (However, based on the results in the main experiment, I strongly suspect that the method proposed in this paper will not be able to achieve the effect of DCPT.)

[1] Residual Denoising Diffusion Models. CVPR 2024.

[2] Selective Hourglass Mapping for Universal Image Restoration Based on Diffusion Model. CVPR 2024.

[3] Universal Image Restoration Pre-training via Degradation Classification. ICLR 2025.

**Methods And Evaluation Criteria:**

Using pre-trained generation models makes sense in the all-in-one image restoration task. The test benchmark selected in this paper is also representative.

**Other Comments Or Suggestions:**

Table 1. Why the `Type` is vertical???

**Other Strengths And Weaknesses:**

**Weakness**

I question the significance of the problem addressed in this paper. ResShift [1], RDDM [2], and DiffUIR [3] have effectively achieved diffusion from low-quality images, providing a better alignment with the physical dynamics of the image restoration process. Furthermore, DiffUIR [3] has also been tested for all-in-one image restoration. The problem tackled in this paper appears somewhat gradual.

[1] ResShift: Efficient Diffusion Model for Image Super-resolution by Residual Shifting. NeurIPS 2023.

[2] Residual Denoising Diffusion Models. CVPR 2024.

[3] Selective Hourglass Mapping for Universal Image Restoration Based on Diffusion Model. CVPR 2024.

**Questions For Authors:**

1. In lines 319-320, this paper claims that "although IRBridge is not explicitly trained for image restoration tasks". However, Sec.4.1 also claims that "degraded images are directly used as conditions for ControlNet". ControlNet is not trained? or ControlNet is not trained for image restoration? What is the function of this ControlNet? (I notice that Appendix D.1 says that "integrate conditional guidance into the pre-trained Stable Diffusion". However, it just trains image restoration. )
2. Appendix D.2 does not describe how the training and test sets of the desnowing dataset are divided.

**Relation To Broader Scientific Literature:**

I believe the method proposed in this paper is helpful for other ill-posed inverse problems.

**Theoretical Claims:**

The proof of Proposition 3.1 is correct in my view.

---

> ### Author Rebuttal · Authors · 2025-04-01
>
> We sincerely appreciate your insightful review and valuable comments! We will carefully address your concerns below.
>
> **Q1: Insufficient comparative experiments.**
>
> We have added a [QUANTITATIVE COMPARISON](https://anonymous.4open.science/r/IRBridge-4181/asserts/table1.md) between IRBridge and the suggested methods across different tasks. Since the training code for DCPT has not been released, we directly used the pretrained model provided by the authors for comparison. All results are reported at a unified resolution of 512×512 to ensure a fair comparison.
>
> The proposed IRBridge outperforms existing methods on most tasks. While it may underperform on deterministic metrics like PSNR and SSIM due to its generative nature and output diversity (e.g., slight tonal shifts from ground truth), these variations rarely affect perceptual quality, though they impact sensitive metrics. In contrast, IRBridge consistently achieves **better FID scores** (lower average ~4.97), highlighting its strength in capturing data distribution. Moreover, we present the quantitative results of IRBridge on real-world datasets in [THIS TABLE](https://anonymous.4open.science/r/IRBridge-4181/asserts/table2.md). IRBridge outperforms the latest method DCPT, achieving an average performance improvement of approximately 5.6%. These results demonstrate the generalization ability of IRBridge.
>
> **Q2: The problem tackled in this paper appears somewhat gradual.**
>
> We thank the reviewer for their thoughtful comments, and we provide insights from both theoretical and practical perspectives.
>
> + **Theoretically.** The mentioned methods, similar to IRSDE,  can be regarded as defining a stochastic differential equation (SDE) that describes the transformation from LQ images to HQ counterparts. They typically train models to estimate the score of the corresponding reverse SDE for image restoration. **In contrast**, our method explores the use of score estimators from *generative SDEs* (pretrained DMs) to estimate the score of these SDEs. By leveraging a generative model capable of estimating $p(x_{0} | x_t^*, x_{lq})$, IRBridge can directly solve the reverse SDE defined by methods like IRSDE and GOUB.
> + **Practically.** Our method is the first to enable bridge methods to directly leverage pretrained DMs, whose priors not only reduce the training burden but also provide better generalization (+5.6% vs DCPT in real-world scenarios) , as demonstrated in **Q1**. Moreover, IRBridge effectively decouples training and inference: by leveraging a DM conditioned on LQ images, IRBridge can solve alternative forms of restorative SDEs, without being constrained to specific diffusion coefficients, enabling greater flexibility.
>
> **Q3: The role of ControlNet and training issues.**
>
> The employed ControlNet integrates guidance information from the degraded image, ensuring that the model is directed to generate the desired x0. From another perspective, ControlNet can indeed be seen as performing implicit image restoration, as it models $p(x_{hq} | x_t^*, x_{lq})$. **However, we emphasize a fundamental distinction:** ControlNet is trained via conditional score matching, following standard diffusion. Our ControlNet is trained to estimate the score of the SDE associated with a *generative diffusion*, whereas restoration bridge methods like GOUB are trained to estimate the score of their *restorative diffusion* . Moreover, these methods also use LQ images as condition for the model to enhance performance.
>
> We clarify that IRBridge does not focus on the training of the model itself, yet we acknowledge that its performance depends on the underlying model. The ControlNet we used is a relatively coarse implementation, and other conditional control methods (like UniCon[1]) remain applicable to IRBridge, for fine-tuning the generative model. Future work can explore more effective conditional guidance strategies and guidance modules tailored for IR tasks.
>
> **Q4: Training an all-in-one image restoration bridge cannot be considered a highlight contribution.**
>
> We agree that training an all-in-one model on multiple degradation datasets can achieve good results. However, we want to clarify that the main contribution of IRBridge lies in enabling the solution of IR bridges through pre-trained DMs for the first time, rather than training an all-in-one model. Typically, training an all-in-one model requires merging multiple datasets to handle diverse image types, leading to high computational costs. In contrast, IRBridge allows direct leverage of generative priors, significantly lowering the training effort (discussed in Section 4.2).
>
> **Q5: Table quality and missing details.**
>
> The desnowing dataset contains 50k samples in both the training and testing sets. We will correct the visual quality of the tables and figures and address the missing details in the revised version. Thanks pointing that out!
>
> [1] A Simple Approach to Unifying Diffusion-based Conditional Generation, ICLR 2025

---

> > ### Comment · Reviewer_1Y3v · 2025-04-03
> >
> > I appreciate the author's rebuttal, which answered many of my questions, but I still have some issues:
> > 1. The results of DiffUIR do not seem to be presented in [QUANTITATIVE COMPARISON](https://anonymous.4open.science/r/IRBridge-4181/asserts/table1.md), please add it in time.
> > 2. I noticed that the author emphasized the superiority of the IRBridge in the FID indicator. But when the number of test samples is small, such as Rain100H has only 100 test images and LoLv1 has only 15 test images, the FID indicator is not accurate enough. On the RESIDE SOTS test set (with nearly 500 test images), the FID of IRBridge is also similar to that of DCPT-PromptIR. It seems that IRBridge has not increased significantly compared with DCPT, but has large-scale parameters of pre-trained generative models.
> > 3. I am very curious about the evaluation of `generalization` mentioned in this rebuttal. The generalization research of image restoration models is very important, which may be further emphasized in the experimental section of this paper.
> >
> > For now, I tend to raise the score to 2.5, which means "but could also be accepted" part of the `Overall Recommendation:`. If the authors address my above issues and modify this paper accordingly, I think this paper will meet the acceptance criteria of the ICML conference.

---

> > > ### Author Response · Authors · 2025-04-03
> > >
> > > We appreciate your positive feedback! Next, we will address your concerns point by point.
> > >
> > > **Q1. The results of DiffUIR.**
> > >
> > > **A1.** We apologize for the delay. We have now included a [QUANTITATIVE COMPARISON](https://anonymous.4open.science/r/IRBridge-4181/asserts/table1.md) with DiffUIR. As shown, IRBridge outperforms DiffUIR across all tasks, with particularly notable improvements in PSNR (+0.606) and FID (−8.872), demonstrating the superiority of our method.
> > >
> > > **Q2.** **Evaluation of IRBridge Using FID.**
> > >
> > > **A2.**  We sincerely appreciate you pointing this out! We acknowledge the limitation of FID when evaluated on small-scale datasets. In response, We have included additional comparison results on the OutdoorRain [1] dataset, a combined low-light enhancement dataset, and a combined dehazing dataset.
> > >
> > > Specifically, for the low-light enhancement task, we combined the test set of LOL v1, the training set of LOL v2 [2], and its test set, resulting in a total of 1,115 samples.  For dehazing, we merged the original SOTS with the Unannotated subset from RESIDE-β (4,808 samples), resulting in a total of 5,308 samples. Since some of the data lacks ground-truth annotations, we computed the FID using the GT images from the OTS dataset. To the best of our knowledge, DCPT has not been pre-trained on OutdoorRain or LOL v2, and neither has our model, which ensures a fair comparison.  The results are as below:
> > >
> > > | Dataset        | OutdourRain (2700 Samples)     | Combined LOL (1115 Samples)    | Combined RESIDE (5308 Samples) |
> > > | -------------- | ------------------------------ | ------------------------------ | ------------------------------ |
> > > | Method         | PSNR / SSIM / LPIPS / FID      | PSNR / SSIM / LPIPS / FID      | FID                            |
> > > | DCPT-Restormer | 26.79 / 0.8198 / 0.161 / 35.85 | 24.01 / 0.8641 / 0.059 / 47.55 | 24.98                          |
> > > | DCPT-PromptIR  | 27.33 / 0.8257 / 0.154 / 30.87 | 23.87 / 0.8597 / 0.062 / 46.33 | 25.64                          |
> > > | IRBridge       | 28.57 / 0.8678 / 0.153 / 24.59 | 25.19 / 0.8967 / 0.058 / 35.87 | 17.23                          |
> > >
> > > On larger-scale datasets, IRBridge achieves better performance  (Avg. PSNR +5.03%, SSIM +4.7%) compared to DCPT, and achieves a more significant FID improvement (-8.41) on the image dehazing task, demonstrating its effectiveness.
> > >
> > > **Q3. Large-scale parameters of pre-trained generative models.**
> > >
> > > **A3.** We acknowledge the increased parameter count introduced by the use of pre-trained DMs, which will be a focus of our future work. However, we would like to emphasize that IRBridge offers significantly more efficient training compared to DCPT. Specifically, our model was trained for 10k iterations with a batch size of 12 on a single 3090 GPU, whereas DCPT was trained for 750k iterations with a batch size of 32 on 4 NVIDIA L40 GPUs. Compared to DCPT, our method is actually much more cost-efficient.
> > >
> > > **Q4. The evaluation of generalization**
> > >
> > > **A4.** We greatly appreciate your constructive feedback! Following your suggestion, we will discuss the generalization capability in the revised `Experiments` section from two perspectives:
> > >
> > > + **Cross-domain Generalization.**  As mentioned in the paper (Section 4.2), models trained solely on a face dataset using IRBridge demonstrate better generalization to other domains (including indoor and outdoor scenes) compared to models trained from scratch. This highlights IRBridge’s ability to leverage pre-trained priors for improved cross-domain performance, which is important for image restoration tasks, as many IR datasets are limited to specific domains.
> > > + **Generalization to Real-world Scenarios.** We will include in the revision a quantitative comparison between IRBridge and other methods (including the recent DCPT) on real-world datasets such as RealRain-1K, RealSnow, and RTTS, to demonstrate the generalization capability of IRBridge in real-world scenarios. Additionally, we will provide visual results in `Appendix E` to offer an intuitive illustration of its performance under real-world conditions.
> > >
> > > [1] Heavy Rain Image Restoration: Integrating Physics Model and Conditional Adversarial Learning
> > >
> > > [2] From Fidelity to Perceptual Quality: A Semi-Supervised Approach for Low-Light Image Enhancement
> > >
> > > **We sincerely thank the reviewer for recognizing our work and for raising the score after our response! Your encouragement means a great deal to us. We also greatly value the comments from the other reviewers and have addressed each of them in the revised manuscript. We hope these improvements will help all reviewers better appreciate the value of our work.**

---

### Official Review · Reviewer_AyHU · 2025-03-13

**Overall Recommendation:** 3

**Summary:**

This paper introduces a new approach for leveraging pre-trained generative diffusion models in image restoration bridges. Traditional image restoration bridge models require training from scratch for each degradation type, making them computationally expensive. This work aims to eliminate that requirement by integrating generative priors into the restoration process.

**Claims And Evidence:**

The claims are generally well-supported by evidence, with mathematical analysis and empirical validation.

**Essential References Not Discussed:**

None.

**Experimental Designs Or Analyses:**

The designs and analyses appear sound, but some evaluations rely on synthetic degradations, which might not capture the full complexity of real-world scenarios.

**Methods And Evaluation Criteria:**

Yes, the proposed methods and evaluation criteria are well-aligned with the problem of image restoration.

**Other Comments Or Suggestions:**

The method relies on manual hyperparameter tuning for timestep scheduling, and further evaluations on real-world degradations would strengthen its practical relevance.

In my view, the main novelty of this paper lies in the transition equation. How does the proposed transition equation differentiate IRBridge from existing image restoration approaches, and in what ways does it offer an advantage over prior diffusion-based restoration methods?

Some typos, for example "Critial timsteps" should be "Critical timesteps" in several figures and sections.

The iterative nature of IRBridge may lead to slower inference, making it essential to quantify its efficiency. Assessing the trade-off between accuracy and speed would offer valuable insights for practitioners when choosing between IRBridge and other restoration methods.

**Other Strengths And Weaknesses:**

It combines ideas from diffusion-based generative modeling and image restoration bridges by introducing a transition equation that leverages pre-trained generative priors.

Its empirical evaluations across multiple restoration tasks demonstrate improved performance and generalization, and the theoretical derivations are presented rigorously.

**Questions For Authors:**

Please see the above comments.

**Relation To Broader Scientific Literature:**

The paper synthesizes and extends prior work in diffusion-based image generation and restoration bridges by introducing a transition equation that leverages pre-trained generative priors—building on foundational ideas from diffusion models (e.g., Sohl-Dickstein et al., Ho et al.) and restoration frameworks (e.g., IR-SDE, GOUB) to eliminate the need for training separate models for each degradation type.

**Theoretical Claims:**

The proof for Proposition 3.1 appears to be correct and complete.

---

> ### Author Rebuttal · Authors · 2025-04-01
>
> We sincerely appreciate your insightful comments! We will address your concerns point by point.
>
> **Q1: Evaluations on real-world scenarios.**
>
> We present the [EVALUATION RESULTS](https://anonymous.4open.science/r/IRBridge-4181/asserts/table2.md) on real-world datasets (RealRain-1K for deraining, RealSnow for desnowing, and RTTS for dehazing). As these datasets lack ground truth, we utilized widely-used no-reference image quality metrics (MUSIQ, BRISQUE, NIQE) to evaluate the performance of the methods. As shown in the table, IRBridge demonstrates a clear advantage over other training-from-scratch methods in real-world scenarios. Compared to the latest method, DCPT (ICLR 2025), IRBridge still achieves a better performance (with average 5.6% performance improved), highlighting its superior performance under real-world conditions. We attribute these improvements to the pretrained generative prior that equips the model with stronger generalization capability. Visual examples are provided [HERE](https://anonymous.4open.science/r/IRBridge-4181/asserts/real/show.md).
>
> **Q2: Inference efficiency.**
>
> We acknowledge that the iterative nature of IRBridge makes it less suitable for real-time processing. However, its high flexibility allows for improving efficiency by reducing the number of inference steps. We present the performance and inference time of IRBridge under different numbers of inference steps (on a 3090 GPU processing standard 512×512 resolution images). The [TABLE](https://anonymous.4open.science/r/IRBridge-4181/asserts/table2.md) shows that even with a 75% reduction in inference steps, the performance degradation remains minimal (< 3%), demonstrating it’s potential for lowering inference overhead.  Additionally, future work could focus on distillation and compression of the pretrained generative model to further reduce the computational complexity of IRBridge and improve its inference efficiency.
>
> **Q3: Manual Hyperparameter Tuning.**
>
> We acknowledge that it is a limitation of IRBridge. To address this, we offer two insights:
>
> + **1) Manual configuration as a practical solution.**  In Appendix C, we show the impact of timestep selection strategies across different tasks. Our experiments indicate that although timestep selection impacts intermediate states, it has a minimal effect (~5%) on final performance, suggesting manual tuning is both sufficient in most scenarios. Based on our experience, the timestep schedule used in inpainting task tends to work well for most tasks, which is referred as the default setting.
>
> + **2) Systematic hyperparameter selection.** Several approaches can provide more principled strategies for selecting optimal parameters under specific considerations. For example, the recent work LDSB [1] proposes using a  path prediction network to estimate more suitable diffusion coefficients via a local Schrödinger bridge, thereby improving the performance of pretrained DMs. Similarly, for IRBridge, one could explore optimal timestep scheduling under the context of minimizing relative entropy.
>
> For a trade-off between cost and performance, we still recommend using manually selected parameters. Pursuing complex optimization for marginal performance gains is not cost-effective.
>
> **Q4: How does the transition equation differentiate IRBridge from other approaches, and in what ways does it offer an advantage over diffusion-based IR methods?**
>
> We would like to elaborate on the contributions of IRBridge from both theoretical and practical perspectives.
>
> + **1) Theoretically.** IRBridge supports bridging two distinct diffusion processes that share the same endpoints. In contrast to previous IR bridge methods, IRBridge does not define a specific diffusion process tailored for restoration. Instead, it leverages the transition equation to ”substitute“ a pretrained DDPM model in place of a specially trained score network for solving the corresponding SDE. To our knowledge, this is the first work to introduce a pretrained generative model into such bridge-based restoration frameworks, breaking the limitation of prior approaches that require training models from scratch.
> +  **2) Practically.** The incorporation of pretrained generative priors endows IRBridge with stronger generalization capabilities, especially in complex scenes or across different data domains. In **Q1**, we demonstrated the superior generalization ability of IRBridge compared to other bridging methods in real-world scenarios. Additionally, as discussed in Section 4.2, the pretrained model provides a strong initialization, enabling IRBridge to achieve nearly twice the training efficiency compared to methods trained from scratch (e.g., GOUB).
>
> **Q5: Correction of some typos.**
>
> We will thoroughly check for spelling errors in the subsequent revision to improve the quality of our manuscript. Your feedback is invaluable for improving our work!
>
> [1] Finding Local Diffusion Schrodinger Bridge using Kolmogorov-Arnold Network, CVPR 2025

---

> > ### Comment · Reviewer_AyHU · 2025-04-07
> >
> > Thanks for the authors' detailed responses. The evaluations in real-world scenarios and the explanation on contributions has addressed my concerns. However, the inference efficiency and the manual hyperparameter tuning remain complex and non-trivial, which I believe are still significant limitations of the work. That said, considering the improvements and clarifications provided, I am willing to raise my score.

---

> > > ### Author Response · Authors · 2025-04-07
> > >
> > > **We sincerely appreciate the reviewer’s positive feedback on our work and the increased recommendation score after our response !**
> > >
> > > We acknowledge that the manual inference hyperparameters currently used in IRBridge are not theoretically optimal. However, we have identified systematic methods to obtain optimal settings under specific considerations. From a cost-effectiveness standpoint, we still advocate for using manually tuned hyperparameters, as they already deliver strong performance with minimal overhead.
> > >
> > > In terms of inference efficiency, diffusion-based methods are indeed less competitive than regression-based models due to their inherently iterative inference paradigm. Nevertheless, IRBridge has demonstrated the ability to reduce the number of inference steps by flexibly adjusting diffusion parameters, enabling a significantly faster inference process with only minimal performance degradation. Moreover, future work can focus on model distillation and compression to further reduce its computational burden.
> > >
> > > **Once again, we truly appreciate your insightful and constructive comments, which have helped us further improve our work. We will incorporate your suggestions and revise the manuscript accordingly in the next version.**

---

### Official Review · Reviewer_S4oL · 2025-03-14

**Overall Recommendation:** 2

**Summary:**

Traditional image restoration bridge models require training from scratch for each degradation type, limiting efficiency and generalization. Meanwhile, pretrained generative diffusion models are underutilized due to mismatched intermediate states between generative and restorative diffusion processes. This work proposes a transition equation that connects two diffusion processes with the same endpoint distribution. Based on the equation, the IRBridge framework is introduced to directly utilize generative models for image restoration tasks.

## update after rebuttal
I appreciate the authors’ detailed response. My main concern regarding the experiments, including the baseline methods and reported performance gains, still remains. Therefore, I maintain my score of 2.

**Claims And Evidence:**

Yes, the claims made in the submission are supported by evidence.

**Essential References Not Discussed:**

Yes, the related works are essential to understand of the context for the key contributions of the paper.

**Experimental Designs Or Analyses:**

Yes, I have reviewed the validity of the experiment design and analysis.

**Methods And Evaluation Criteria:**

Overall, the proposed method and evaluation criteria, including the benchmark dataset, are reasonable. However, the size of the test data is not specified. Additionally, based on the images in the paper, it appears that the method only supports fixed-size inputs. The authors should provide more detailed explanations on this aspect.

**Other Comments Or Suggestions:**

N/A

**Other Strengths And Weaknesses:**

Strengths:
1. The proposed transition equation concept remove restrictive assumptions in prior work. Eliminating per-degradation training aligns with real-world needs for scalable solutions, making the approach both practical and original in application design.
2. Demonstrates state-of-the-art performance on 6 restoration tasks, with potential for immediate use in photography and autonomous vehicles.
3. The paper is well-structured, with method descriptions presented in a logical and layered manner.

Weaknesses:
1. The method is based on IR-SDE and GOUB, with pretrained diffusion priors introduced to enhance performance. However, the performance gains from these priors are unclear. Results on some datasets are missing, and the performance improvement in Deraining and Image Inpainting is very limited.
2. The authors should provide the resolution of the test images and the inference efficiency, as these are important for image restoration tasks.
3. Although the authors have tested six image restoration tasks, the comparisons on these tasks are not sufficiently comprehensive. The authors should compare their method with the latest works.

**Questions For Authors:**

Please see the weaknesses.

**Relation To Broader Scientific Literature:**

The core contribution of the paper further enhances the effectiveness of using pretrained diffusion models to complete image restoration.

**Theoretical Claims:**

Yes, I checked the correctness of the theoretical proofs presented in the paper. I did not identify significant errors.

---

> ### Author Rebuttal · Authors · 2025-04-01
>
> We are grateful for your valuable comments, which are helpful for improving our manuscripts. Below, we will address your concerns point by point.
>
> **Q1: Test image resolution and supported resolution.**
>
> To ensure a fair comparison, we conducted all comparison experiments at a 512×512 resolution in our paper. However, it should note that the resolution supported by IRBridge depends on the model it employs. For SD1.5’s UNet, it supports resolutions in multiples of 64. Inspired by WeatherDiff, we developed an overlapping patches partitioning scheme to support inputs of arbitrary resolutions. We present [VISUAL EXAMPLES](https://anonymous.4open.science/r/IRBridge-4181/asserts/real/show.md) of IRBridge processing arbitrarily high-resolution real-world images, highlighting its ability to effectively handle high-resolution images without causing performance degradation.
>
> **Q2: Inference efficiency of IRBridge.**
>
> With the default setting of 100 inference steps, IRBridge takes about 14.2 seconds to process a 512 resolution image, running at 7.18 steps per second. Notably, since IRBridge decouples model training from inference, it allows flexible specification of both the diffusion coefficients and the number of inference steps. In [THIS TABLE](https://anonymous.4open.science/r/IRBridge-4181/asserts/table3.md) , we present the inference time and quantitative results of IRBridge under different step settings. It is worth noting that even with a 75% reduction in inference steps (14.2s -> 3.4s of inference time), IRBridge still maintains comparable performance (<3% average drop), demonstrating its potential for more efficient inference.
>
> **Q3: The performance gains from pretrained diffusion priors.**
>
> The performance gains brought by pretrained priors can be attributed to three main aspects:
>
> + **1) Improved Generalization.** Pretrained models provide general knowledge about a wide range of image distributions, enhancing the model’s ability to generalize to various scenarios. IRBridge demonstrates better generalization across different data domains compared to methods trained from scratch (as shown in Fig.7).  Furthermore, we present the results of IBridge on real-world datasets (see **Q4**), showing its superior performance compared to other methods, further supporting our claims.
> + **2) Faster Convergence.** As shown in Section 4.2, IRBridge benefits from a favorable initialization provided by the pretrained model, leading to nearly twice the training efficiency compared to training-from-scratch methods such as GOUB.
> + **3) Better Robustness.** Even though we simply adopted a ControlNet trained with a DDPM objective, its performance still outperforms methods trained specifically to estimate the score of a particular SDE. This demonstrates that the pretrained generative prior provides IRBridge with more robust representations, thereby enhancing its overall performance.
>
> **Q4: Comparison on more datasets and with the latest methods.**
>
> We provide [QUANTITATIVE RESULTS](https://anonymous.4open.science/r/IRBridge-4181/asserts/table1.md) of IRBridge compared with recent methods, including ResShift [1], RDDM [2], DiffUIR [3], and DCPT [4], across different tasks. Additionally, we present the [QUANTITATIVE RESULTS](https://anonymous.4open.science/r/IRBridge-4181/asserts/table2.md) of IRBridge and the aforementioned methods on real-world degraded datasets, covering image deraining, desnowing, and dehazing.
>
> IRBridge outperforms other bridging methods, achieving an average improvement of approximately 3.23% over the latest bridge method DiffUIR, demonstrating that the pretrained generative prior significantly enhances model robustness. Furthermore, IRBridge achieves an overall improvement of about 5.60% over the recent method DCPT on real-world scenarios, confirming that the introduced prior in IRBridge improves the model's generalization across different degradation conditions. We present [VISUAL EXAMPLES](https://anonymous.4open.science/r/IRBridge-4181/asserts/real/show.md) of IRBridge processing real-world degraded images.
>
>  **Q5: The size of the test data is not specified.**
>
> We present the complete number of training and testing samples for the datasets used in the table below.
>
> | Datasets | Training samples | Test Samples |
> | - | - | - |
> | Rain100H | 1800 | 100 |
> | RESIDE | 72,135 | 4,322 |
> | Snow100K | 50,000 | 50,000 |
> | Raindrop | 861 | 58 |
> | LOL | 485 | 15 |
> | CelabA-HQ | 20,000 | 20,000 |
>
> We thank the reviewer for the thorough review. These detailed suggestions are invaluable for improving the quality of our manuscript.
>
> [1] ResShift: Efficient Diffusion Model for Image Super-resolution by Residual Shifting. NeurIPS 2023
>
> [2] Residual Denoising Diffusion Models. CVPR 2024
>
> [3] Selective Hourglass Mapping for Universal Image Restoration Based on Diffusion Model. CVPR 2024
>
> [4] Universal Image Restoration Pre-training via Degradation Classification. ICLR 2025.

---

> > ### Comment · Reviewer_S4oL · 2025-04-09
> >
> > Thank you for the detailed responses, which have addressed most of my concerns. In the initial version, the method was compared with only two bridging models per task, which I found insufficient. Although the authors have added several recent methods for comparison, the performance improvements in some tasks—such as SSIM and LPIPS on Rain100H, Snow100K, and RESIDE—remain marginal. Taking into account the completeness of the experiments, the method’s performance and efficiency, as well as the evaluations from other reviewers, I decided to maintain my original score.

---

> > > ### Author Response · Authors · 2025-04-09
> > >
> > > Thanks for your valuable feedback. We would like to reiterate and clarify several important points.
> > >
> > > **Q1: The performance improvement.**
> > >
> > > **A1.** So far, we have conducted comprehensive comparisons with other bridge methods, including IRSDE (ICML 23), GOUB (ICML 24), ResShift (NeurIPS 23), RDDM (CVPR 24), and DiffUIR (CVPR 24). We clarify that IRBridge achieves consistent performance advantages over the latest bridge method DiffUIR across all tasks, with an average improvement of approximately:
> > >
> > > - **PSNR↑**: +2.37%
> > > - **SSIM↑**: +0.79%
> > > - **LPIPS↓**: −7.86%
> > > - **FID↓**: −30.39%
> > >
> > > Even compared to the recent DCPT (ICLR 2025), IRBridge achieves an average performance gain of nearly **5.6%** in real-world scenarios. These quantitative results strongly demonstrate the performance improvement brought by introducing pretrained generative priors in IRBridge.
> > >
> > > In addition, we would like to clarify the significance of the performance improvements. According to the results reported in the original papers:
> > >
> > > >  1. **DiffUIR-L (CVPR 24)** improves upon **RDDM (CVPR 24)** across all tasks by (avg. PSNR +0.818 / 3.05%, SSIM +0.0 / 0.0%)
> > > >
> > > >  2. **GOUB-SDE (ICML 24)** improves upon **IRSDE (ICML 23)** on the image deraining task by (PSNR +0.31 / 0.97%, SSIM -0.0013 / -1.41%, LPIPS -0.001 / -2.12%,  FID -0.5 / -2.68%).
> > >
> > > **Therefore, the performance gain of our method is in fact noticeable.**
> > >
> > > **Q2:  Efficiency.**
> > >
> > > **A2 (1) Training Efficiency.** Benefiting from the generative prior, IRBridge achieves significantly higher training efficiency.
> > >
> > > + Compared to the recent pretrained method DCPT, IRBridge requires only 10K training iterations with a batch size of 12 on a single RTX 3090 GPU, whereas DCPT is trained for 750K iterations on 4×L40 GPUs.
> > > + Compared to bridge models trained from scratch, IRBridge still demonstrates superior efficiency—for instance, its actual training time is less than 50% of that of GOUB (1 days 3090 GPU vs. 2.5 days 3090 GPU), while achieving better performance.
> > >
> > > **A2 (2) Inference Efficiency.**
> > >
> > > Although IRBridge incorporates  a pretrained model leading to a relatively larger parameter count, it allows for flexible adjustment of diffusion process during inference, **which enables the use of fewer sampling steps to improve inference efficiency.**  In contrast to other bridge methods that typically adopt relatively fixed iteration steps, IRBridge can achieve shorter inference time in practice. We present a comparison of inference efficiency with GOUB on a single RTX 3090 GPU in the table below:
> > >
> > > | Method   | Inference Speed     | Inference Steps | Inference Time |
> > > | -------- | ------------------- | --------------- | -------------- |
> > > | GOUB     | **8.10 iter / sec** | 100             | 11.3s          |
> > > | IRBridge | 7.18 iter / sec     | **25**          | **3.4s**       |
> > >
> > > While each inference step in IRBridge takes longer, **the reduced number of steps leads to more efficient overall inference (11.3s -> 3.4s).**
> > >
> > > Based on the above rebuttal, we believe your concerns have been substantively addressed. **Finally, we still hope you would kindly reconsider your score.** We would like to once again express our sincere thanks for your time and effort in reviewing our work. Your comments have been highly constructive and instrumental in improving the quality of our manuscript.

---

### Decision · Program_Chairs · 2025-05-01

**Decision:**

Accept (poster)

**Comment:**

In order to better utilize generative models within image restoration bridges, this paper proposes a transition equation that bridges two diffusion processes with the same end point distribution. The paper originally received 3xWeakReject and 1xWeakAccept. The main concerns include lack of discussions on inference efficiency, insufficient evaluations, manual hyperparameter tuning, missing details, etc. The authors have provided rebuttals and addressed most concerns of reviewers. Afterward, two reviewers raise their ratings. In the discussion phase, Reviewer S4oL still keeps negative about the performance and efficiency of the proposed method. The authors have clarified these concerns in their second rebuttal. Considering the rebuttal and discussions from all reviewers, ACs recommend accepting this paper. The authors are suggested to carefully revise the paper and incorporate newly conducted experiments according to the comments and discussions.